# The hydrological cycle and ocean circulation of the Maritime Continent in the Pliocene: results from PlioMIP2

Xin Ren[1], Daniel J. Lunt[1], Erica Hendy[2], Anna von der Heydt[3], Ayako Abe-Ouchi[4], Bette Otto-Bliesner[5], Charles J. R. Williams[1,6], Christian Stepanek[7], Chuncheng Guo[8], Deepak Chandan[9], Gerrit Lohmann[7], Julia C. Tindall[10], Linda E. Sohl[11,12], Mark A. Chandler[11,12], Masa Kageyama[13], Michiel L. J. Baatsen[3], Ning Tan[14,15], Qiong Zhang[16], Ran Feng[17], Stephen Hunter[10], Wing-Le Chan[4], W. Richard Peltier[9], Xiangyu Li[18,19,20], Youichi Kamae[21], Zhongshi Zhang[8,19], and Alan M. Haywood[10]

[1]School of Geographical Sciences, University of Bristol, Bristol, UK
[2]School of Earth Sciences, University of Bristol, Bristol, UK
[3]Institute for Marine and Atmospheric research Utrecht (IMAU), Department of Physics, Utrecht University, Utrecht, the Netherlands
[4]Atmosphere and Ocean Research Institute, The University of Tokyo, Kashiwa, Japan
[5]Climate and Global Change Dynamics lab, National Center for Atmospheric research, USA
[6]NCAS, Department of Meteorology, University of Reading, Reading, UK
[7]Alfred-Wegener-Institut – Helmholtz-Zentrum für Polar and Meeresforschung (AWI), Bremerhaven, Germany
[8]NORCE Norwegian Research Centre, Bjerknes Centre for Climate Research, Bergen, Norway
[9]Department of Physics, University of Toronto, Toronto, Canada
[10]School of Earth and Environment, University of Leeds, Woodhouse Lane, Leeds, West Yorkshire, UK
[11]Center for Climate Systems Research at Columbia University, New York, NY, USA
[12]NASA Goddard Institute for Space Studies, New York, NY, USA)
[13]LSCE/IPSL – Laboratoire des Sciences du Climat et de l'Environnement, UMR8212, CEA-CNRS-UVSQ – CE Saclay, L'Orme des Merisiers, Gif-sur-Yvette Cedex, France
[14]Key Laboratory of Cenozoic Geology and Environment, Institute of Geology and Geophysics, Chinese Academy of Sciences, Beijing, China
[15]Laboratoire des Sciences du Climat et de l'Environnement, LSCE/IPSL, CEA-CNRS-UVSQ, Université Paris-Saclay, Gif-sur-Yvette, France
[16]Department of Physical Geography and Bolin Centre for Climate Research, Stockholm University, Stockholm, Sweden
[17]Department of Earth Sciences, College of Liberal Arts and Sciences, University of Connecticut, Storrs, USA
[18]Centre for Severe Weather and Climate and Hydro-geological HazardsWuhan, China
[19]Department of Atmospheric Science, School of Environmental Studies, China University of Geosciences, Wuhan, China
[20]Climate Change Research Center, Institute of Atmospheric Physics, Chinese Academy of Sciences, Beijing, China
[21]Faculty of Life and Environmental Sciences, University of Tsukuba, Tsukuba, Japan

**Correspondence:** Xin Ren (xinxin.ren@bristol.ac.uk)

**Abstract.** The Maritime Continent (MC) forms the western boundary of the tropical Pacific Ocean, and relatively small changes in this region can impact the climate locally and remotely. In the mid-Piacenzian warm period of the Pliocene (mPWP; 3.264 to 3.025 million years ago) atmospheric $CO_2$ concentrations were $\sim$400 ppm, and the subaerial Sunda and Sahul shelves made the land-sea distribution of the MC different to today. Topographic changes and elevated levels of $CO_2$, combined with other forcings, are therefore expected to have driven a substantial climate signal in the MC region at this time. By using the results from the Pliocene Model Intercomparison Project phase 2 (PlioMIP2) we study the mean climatic features of the MC

in the mPWP and changes in Indonesian Throughflow (ITF) with respect to preindustrial. Results show a warmer and wetter mPWP climate of the MC and lower sea surface salinity in the surrounding ocean compared with preindustrial. Furthermore, we quantify the volume transfer through the ITF; although the ITF may be expected to be hindered by the subaerial shelves, 10 out of 15 models show an increased volume transport compared with preindustrial.

In order to avoid undue influence from closely-related models that are present in the PlioMIP2 ensemble, we introduce a new metric - the multi-cluster mean (MCM), based on cluster analysis of the individual models. We study the effect that the choice of MCM versus the more traditional analysis of multi-model mean (MMM) and individual models has on the discrepancy between model results and data. We find that models, which reproduce modern MC climate well, are not always good at simulating the mPWP climate anomaly of the MC. By comparing with individual models, the MMM and MCM reproduce the preindustrial sea surface temperature (SST) of the reanalysis better than most individual models, and produces less discrepancy with reconstructed sea surface temperature anomalies (SSTA) than most individual models in the MC. In addition, the clusters reveal spatial signals that are not captured by the MMM, so that the MCM provides us with a new way to explore the results from model ensembles that include similar models.

## 1  Introduction

The Maritime Continent (MC; 10°S–20°N and 90°E–150°E) consists of more than 22,000 islands and lies in the warmest ocean region, the West Pacific Warm Pool (WPWP) (Ramage, 1968; Yoneyama and Zhang, 2020). The near-surface air temperature in most areas of the MC is higher than 27°C throughout the year (Li et al., 2018a). High sea surface temperature (SST) leads to substantial moisture flux into the atmosphere. High levels of moisture and energy characterize the unstable climate in this region and make it "the world's strongest atmospheric convection center" (Yoneyama and Zhang, 2020). Based on satellite data it has been established that the MC is the region that produces the largest amount of precipitation in the world (Adler et al., 2017; Yamanaka et al., 2018). A large amount of rainfall releases vast quantities of latent heat into the atmosphere (1 mm/y equals to  0.08 W/m$^2$), which is an important driver of global atmospheric circulation (Trenberth et al., 2009; Yamanaka et al., 2018). According to these characteristics, the MC has been recognized as the "boiler box" of the global climate system (Ramage, 1968; Neale and Slingo, 2003).

The MC also impacts the oceanic hydrological system in that it links the Pacific Ocean and the Indian Ocean via the Indonesian Throughflow (ITF), which is a key component of global ocean circulation. In general, the ITF transports warm and comparably fresh water from the Pacific Ocean into the Indian Ocean. Dramatic tectonic processes and sea level changes have shaped this gateway that plays an important role in influencing climate both locally and in other regions of the world (Gordon, 2005b; Tillinger, 2011; Yoneyama and Zhang, 2020). According to the International Nusantara Stratifcation and Transport Program (INSTANT), from 2004 to 2006 (Gordon et al., 2010; Sprintall et al., 2009) the ITF transported about 15 Sv of water from the Pacific Ocean into the Indian Ocean; the heat flux (i.e. the effective temperature of the total throughflow) of the total ITF export was 17.6 °C during these three years (Sprintall et al., 2009).

The MC acts as a source of sensible heat for the atmosphere over land, and the geographical configuration of the MC redistributes heat in the western Pacific and the Indian Ocean; both the effect of sensible heat and the geography of the MC are crucial for the onset of the Walker Circulation (Dayem et al., 2007). The variation of the Walker Circulation leads to periodic phenomena that modulate the climate of the MC, including the El Niño–Southern Oscillation (ENSO), the Indian Ocean Dipole (IOD) and the Pacific Decadal Oscillation (PDO). The MC forms the western boundary of the ENSO phenomenon. On the interannual scale, ENSO and IOD play important roles in modulating rainfall and the ITF transport by affecting zonal winds (Wang, 2019). During El Niño years, the easterly trade winds in the Pacific are weakened. As a result, rainfall is decreased and the ITF is weakened (Zhang et al., 2016; Yamanaka et al., 2018) with a peak-to-trough amplitude of about 5 Sv (Feng et al., 2018). During La Niña years, the situation is reversed. The positive phase of IOD normally occurs during the autumn of an El Niño developing year, which offsets the effects of El Niño on the ITF (Wang, 2019; Sprintall and Révelard, 2014). On decadal timescales, the PDO plays a dominant role in regulating the ITF (Wang, 2019; Li et al., 2018b). Similar to the effect of ENSO, in the warm phase of the PDO easterly trade winds become weaker, causing the ITF transport to weaken, and vice versa (Wang, 2019). A modelling study from Tan et al. (2022) highlighted the importance of the geometry of the MC, and found that the shallow opening of the MC can trigger an active Pacific meridional overturning circulation and enhance upwelling strength along the equator from the Central to Eastern Pacific.

It is possible to obtain an "atlas" of the future warming world from climate model simulations (e.g. Iturbide et al. (2021); Gutiérrez et al. (2021)). However, there is no direct evidence for us to validate how well climate models simulate the future. The mid-Piacenzian warm period (mPWP) is an interval in the Pliocene between 3.264 and 3.025 million years (Ma) ago, which has also been described as mid-Pliocene warm period in previous publications (Dowsett et al., 2016). In this study, we use mPWP as the abbreviation for this warm period. According to the Intergovernmental Panel on Climate Change (IPCC) Sixth Assessment Report (AR6) Working Group I (WGI) Technical Summary (Arias et al., 2021), the mPWP is *very likely* 2.5°C to 4.0°C warmer than the preindustrial, and the concentration of $CO_2$ during this period is *very likely* in the range from 360 ppmv to 420 ppmv, which is substantially higher than for the preindustrial period (280 ppmv). In terms of temperature and geography, the Earth System of the mPWP is similar to projections for the end of the 21st century (Dowsett et al., 2016). Consequently, the mPWP is considered one of the most recent analogues to future warm conditions on a broad scale perspective (Salzmann et al., 2009; Burke et al., 2018).

During the Pliocene Epoch, the Indonesian Gateway reorganized due to tectonic processes, and the ITF was restricted progressively (Karas et al., 2009; Auer et al., 2019). This process contributed to variations of climate throughout the Pliocene. For example, during the mid to late Piacenzian (3.3 Ma –2.6 Ma BP), the reorganization of the ITF led to a strengthening of the South Asian Summer Monsoon and acted as a precursor to east African aridification (Cane and Molnar, 2001; Sarathchandraprasad et al., 2021). By using model results, we can explore the mPWP climate in this region. This effort is supported by geological records that provide a variety of evidence towards reconstructing the climate over the MC during the Pliocene. For example, deep-sea deposit samples of the Pliocene can be utilized to reconstruct SST (Wara et al., 2005; Zhang et al., 2014; O'Brien et al., 2014; Fedorov et al., 2015; McClymont et al., 2020). By comparing results of simulations and geological

records, we can validate climate models and gain confidence in projections of future climate if the models perform well for past warm climates.

The Pliocene has been broadly studied; previous work by the Pliocene Research Interpretation and Synoptic Mapping (PRISM; e.g., Dowsett et al. (2013, 2016)) and the Pliocene Model Intercomparison Project (PlioMIP; e.g, Haywood et al. (2010, 2016a, 2020)) provide us with a large amount of paleoclimate reconstructions and simulations of environmental and climatic information. The second phase of PlioMIP (PlioMIP2), that this study relates to, is briefly described in section 2 of this paper. Large-scale climate features of the mPWP as derived from PlioMIP1 and PlioMIP2 are described in Haywood

et al. (2013) and Haywood et al. (2020): on a global scale simulated annual averaged surface air temperature is elevated, with models suggesting an anomaly above the respective conditions of the preindustrial that ranges between 1.8 and 3.6°C based on PlioMIP1 (coupled atmosphere–ocean climate models only) and between 1.7 and 5.2°C based on PlioMIP2; annual mean total precipitation rates are higher by 0.09-0.18 mm d$^{-1}$ (PlioMIP1; coupled atmosphere–ocean climate models only) and 0.07-0.37 mm d$^{-1}$ (PlioMIP2) in the mPWP. All the models from PlioMIP1 and PlioMIP2 simulated a clear polar amplification of the

warming in the mPWP and show that the warming magnitude in the MC is less than the global average. Regarding precipitation changes in the region, the PlioMIP1 ensemble shows an increase over most parts of the MC, especially in the tropics (Fig. 3b in Haywood et al. (2013)). In contrast to PlioMIP1, PlioMIP2 shows spatially heterogeneous changes in precipitation over the MC (Fig. 5b in Haywood et al. (2020)). Moreover, there is an increase in precipitation minus evaporation over exposed continents across this region (Fig. 1 in Feng et al. (2022)). PlioMIP1 studies have shown that tropical overturning circulation

slows down in the mPWP due to the changes in SST, and that the ascending branch of the Walker Circulation over the MC weakens (Corvec and Fletcher, 2017). In terms of climate variability, ENSO exists in the modelled mPWP, but the amplitude of ENSO is weaker than in the preindustrial. This inference is consistent from PlioMIP1 (Brierley, 2015) to PlioMIP2 (Oldeman et al., 2021). However, the mean zonal SST gradient does not decrease consistently across models in PlioMIP2 (Oldeman et al., 2021). Although the MC is important for regional and large-scale climate, there are relatively few studies of the mPWP

focusing on the MC (e.g. Smith et al. (2020)), especially studies that use the state-of-the-art models.

In this paper, we investigate the climate of the MC in the mPWP by employing PlioMIP2 simulations. We study large-scale climate patterns (SST, the net fresh water flux at the surface (precipitation minus evaporation; P-E), salinity at the sea surface (SOS) and wind stress) of the MC and investigate ocean circulation and strength of the ITF in the mPWP. Moreover, we assess the ability of the PlioMIP2 models to simulate the mPWP climate of the MC by cross-validating simulations and models with

proxy reconstructions and reanalysis data. We investigate differences in predictions by individual models with a hierarchical clustering method.

The rest of the paper is organized as follows. Section 2 describes the PlioMIP2 models, experiment design and simulation data that are relevant to this study. Section 3 addresses the following 4 main questions:

  – Q1. How do PlioMIP2 models perform in simulating the mPWP and preindustrial climate of the MC? (Section 3.1);

– Q2. What are the characteristics of the climate (SST, P-E, SOS and wind stress) of the MC in the mPWP in the PlioMIP2 simulations? (Section 3.2);

– Q3. Did the volume transport via the ITF intensify in the mPWP, and what were the characteristics of this throughflow? (Section 3.3);

– Q4. How can we best emphasize results from individual models that may otherwise disappear in the multimodel ensemble mean (MMM)? How can we account for the effect of any bias duplication caused by models from the same 'family' within our analysis? (Section 3.4)

Finally, we discuss drivers for changes of the ITF and of the climate of the MC (Section 4.1); we summarise performance of individual models and of the MMM in simulating mPWP and pre-industrial climate (Section 4.2); and we address similarities and differences between results by individual models (Section 4.3).

## 2 PlioMIP2

In this section, we describe the PlioMIP2 models, experiments and the PlioMIP2 project. In section 2.1, we introduce the PlioMIP2 models and the development of models which participated in the PlioMIP phase 1 and phase 2. Then, the two core experiments of PlioMIP2 adopted in this study are described in terms of boundary conditions (section 2.2).

### 2.1 PlioMIP2 Models

PlioMIP (Haywood et al., 2011) is a project aimed at studying climate and environments of the late Pliocene. PlioMIP is part of the Palaeoclimate Modelling Intercomparison Project (PMIP). In consideration of the development of models and proxy data in the past years, this project has progressed to the second phase (PlioMIP2) (Haywood et al., 2016b, 2020). In this on-going phase, there are more models (17 models in comparison to 11 models that participated in PlioMIP phase 1) (Table 1). Most of the newly participating models are descendants of existing models. For example, CCSM4 participated in PlioMIP1. The models derived from it, CCSM4-Utr, CCSM4-UoT, CESM1.2 and CESM2, participated in the second phase; IPSLCM5A from the Institute Pierre-Simon Laplace (IPSL) participated in PlioMIP, and then joined PlioMIP2 with IPSLCM5A2 and IPSL-CM6A-LR; NorESM-L, which has been developed from CCSM4, participated in the first phase, and then participated in the second phase with NorESM1-F. The atmosphere-only model HadAM3 participated in PlioMIP1 as a component of the UK Hadley Centre atmosphere-ocean general circulation model HadCM3; in PlioMIP2 it contributed via the "CMIP6-class" UK Met Office HadGEM3-GC31-LL (Williams et al., 2021), as well as via HadCM3 (Hunter et al., 2019). In terms of the atmospheric and oceanic components, new model versions were developed in that the parameterization changed (CCSM4-UoT (Peltier and Vettoretti, 2014) and CCSM4-Utr use a simplified version of ocean mixing scheme from CCSM4 with parameterization changes; NorESM1-F updated the parameterization from NorESM-L), model components were updated (CESM1.2 and CESM2 adopted the atmospheric components of CAM5 and CAM6), or the resolution has been enhanced (IPSL models; NorESM1-F applied the tripolar grid in the ocean component compared with NorESM-L). All these models have run the core *mid-Pliocene-eoi400* experiment (Eoi400) and provide the preindustrial experiment (E280) as a control simulation. For more details on the 16 PlioMIP2 models, see Haywood et al. (2020).

Table 1: Details of the 16 PlioMIP2 models (Haywood et al., 2020) and of the HadGEM3 model used in this study.

| Model ID | modelling centre responsible for simulation | Atmosphere component, resolution, and layers | Ocean component, resolution, and layers | equilibrium climate sensitivity (ECS) | PlioMIP2 Publication |
|---|---|---|---|---|---|
| CCSM4 | National Centre for Atmospheric Research, US | CAM4 FV0.9°×1.25° L26 (Neale et al., 2010) | Parallel Ocean Program version 2 (POP2) G16 ($\sim 1°$) L60 (Danabasoglu et al., 2012) | 3.2 | (Feng et al., 2020a) |
| CCSM4-Utr | Utrecht University, the Netherlands | CAM4 FV2.5°× 1.9° L26 | POP2 with parameterization changes G16 ($\sim 1°$) L60 | 3.2 | (Baatsen et al., 2022) |
| CCSM4-UoT | University of Toronto, Canada | CAM4 FV0.9°×1.25° L26 | POP2 with parameterization changes G16 ($\sim 1°$) L60 | 3.2 | (Chandan and Peltier, 2017) |
| CESM1.2 | National Centre for Atmospheric Research, US | CAM5 FV0.9°×1.25° L30 (Neale et al., 2010) | POP2 G16 ($\sim 1°$) L60 | 4.1 | (Feng et al., 2020a) |
| CESM2 | National Centre for Atmospheric Research, US | CAM6 FV0.9°×1.25° L32 (Danabasoglu et al., 2020) | POP2 with update on parameterization and schemes G16 ($\sim 1°$) L60 (Danabasoglu et al., 2020) | 5.3 | (Feng et al., 2020a) |
| COSMOS | Alfred Wegener Institute, Germany | ECHAM5 T31 (3.75°× 3.75°) L19 (Roeckner et al., 2003) | MPI-OM GR30 (3.0°× 1.8°) L40 (Marsland et al., 2003) | 4.7 | (Stepanek et al., 2020) |

| | | | | | |
|---|---|---|---|---|---|
| EC-Earth3.3 | Stockholm University, Sweden | IFS cycle 36r4 T159 ( ~1.125°× 1.125°) L62 | NEMO3.3 ORAC1 (1°× 1°) L46 (Madec, 2012) | 4.3 | (Zhang et al., 2021a) |
| GISS-E2-1-G | Goddard Institute for Space Studies, US | 2.0°× 2.5° L40 (Kelley et al., 2020) | GISS Ocean v1 1.0°× 1.25° L40 (Kelley et al., 2020) | 3.3 | n.a. |
| HadCM3 | University of Leeds, UK | 2.5°× 3.75° L19 | 1.25°× 1.25° L20 | 3.5 | (Hunter et al., 2019) |
| HadGEM3 | University of Bristol, UK | N96 (1.875°× 1.25°) L85 | NEMO 3.6 1.0°× 1.0°L75 | 5.55 | (Williams et al., 2021) |
| IPSLCM5A | Laboratoire des Sciences du Climat et de l'Environnement, France | LMDZ5 revision 2063 3.75°× 1.875° L39 (Hourdin et al., 2013) | NEMOv3.2 2.0°× 2.0°, 0.5°in the tropics L31 (Madec, 2012) | 4.1 | (Tan et al., 2020) |
| IPSLCM5A2 | Laboratoire des Sciences du Climat et de l'Environnement, France | LMDZ5 revision 3342 3.75°× 1.875° L39 (Hourdin et al., 2013; Sepulchre et al., 2020) | NEMOv3.6 2.0°× 2.0°, 0.5°in the tropics L31 (Sepulchre et al., 2020) | 3.6 | (Tan et al., 2020) |
| IPSL-CM6A-LR | Laboratoire des Sciences du Climat et de l'Environnement, France | LMDZ6A-LR 2.5°× 1.26° L79 (Hourdin et al., 2020; Boucher et al., 2020) | NEMOv3.6 ~1°, latitudinal refined at 1/3°in the equatorial region L75 (Hourdin et al., 2020) | 4.8 | n.a. |
| MIROC4m | University of Tokyo, Japan | T42 (~2.8°× 2.8°) L20 | 1.4°longitude, 0.56°-1.4°latitude L43 | 3.9 | (Chan and Abe-Ouchi, 2020) |
| MRI-CGCM2.3 | University of Tsukuba, Japan | T42 (~2.8°× 2.8°) L30 | 2.5°longitude, 2.0 - 0.5°latitude L26 | 2.8 | (Kamae et al., 2016) |

| | | | | | |
|---|---|---|---|---|---|
| NorESM-L | NORCE Norwegian Research Centre, Bjerknes Centre for Climate Research, Norway | CAM4 Oslo version T31 (∼3.75°× 3.75°) L26 | Miami Isopycnic Co-ordinate Ocean Model (MICOM)<br>G37 (∼3.0°× 3.0°) L30 | 3.1 | (Li et al., 2020) |
| NorESM1-F | NORCE Norwegian Research Centre, Bjerknes Centre for Climate Research, Norway | CAM4 Oslo version 1.9°× 2.5° L26 | MICOM<br>∼1.0°× 1.0°L53 | 2.3 | (Li et al., 2020) |

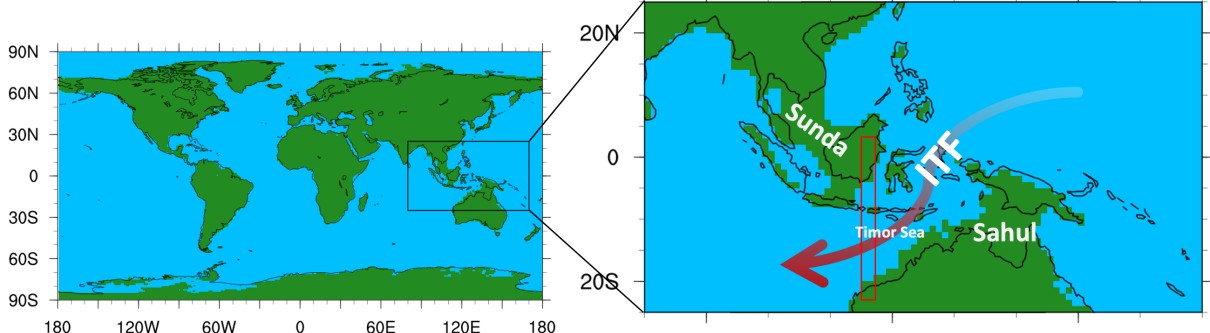

**Figure 1.** PRISM4 Pliocene land-sea mask. The modern continental outline is indicated with black lines; blue and green shadings indicate ocean and land in the Eoi400 experiment. The red rectangle indicates the location of the Timor passage of the Indonesian Throughflow (ITF) that we refer to in section 3.3. The red arrow denotes the position and direction of the ITF illustratively. The Sunda and Sahul shelves are illustrated in this figure via white labels. The continental outline is constructed from Dowsett et al. (2016) and retrodicted with a set of procedures based on the paleogeographic maps of Markwick (2007).

## 2.2 PlioMIP2 Experimental Design

The global-scale and gridded paleo reconstruction data from PRISM provides data for mPWP paleo-geography and land ice,
etc., to enable preparation of reliable boundary conditions and to carry out climate simulations of the mPWP. For PlioMIP phase
1, experiment boundary conditions for the mPWP have been built based on PRISM3D (Dowsett et al., 2010). In PlioMIP2, new
boundary conditions have been adopted, which are derived from the state-of-the-art reconstruction PRISM4 (Dowsett et al.,
2016).

In this study we adopt the E280 simulation as a control experiment, which is forced with preindustrial conditions, and the
Eoi400 simulation as a sensitivity experiment for mPWP conditions, which is forced with all mPWP boundary conditions; these
are topography, ice sheets, soil, vegetation, lakes, land-sea mask and a concentration of 400 ppm of $CO_2$ in the atmosphere.
Simulation naming follows the rules established by Haywood et al. (2016a). The numbers 280 and 400 indicate the $CO_2$
concentration; character o indicates changes to orography, bathymetry, land–sea mask, vegetation, lakes, and soils from the
preindustrial model setup towards mPWP conditions in regions free of ice sheets today, character i indicates such changes
in regions of current ice sheets. Fig. 1 shows the land-sea masks used in the E280 and the Eoi400 experiments indicated by
black lines and colours, respectively, and illustrates the location of the MC. From this map it is clear that in the mPWP, several
modern ocean gateways were absent, such as the Bering Strait, the water body connecting Canada and Greenland (Canadian
Arctic Archipelago), and some straits in the MC. Details of the experimental design are described by Haywood et al. (2016a).
The initial conditions of ocean salinity are either derived from Levitus and Boyer (1994), from an equilibrium state of the
modern (control) simulation, or from the end of the PlioMIP1 experiment (Haywood et al., 2011). Note that HadGEM3 did
not change the land sea mask in the Eoi400 experiment, and therefore for this model the various gateways are the same as
in E280. Data from the experiments E280 and Eoi400 used in this study are from the PlioMIP2 dataset, which is available
at https://esgf-node.llnl.gov/search/cmip6/ and https://geology.er.usgs.gov/egpsc/prism/7.2_pliomip2_data.html. In this study,
the results are calculated across the final 100 model years of each simulation. Data has been regridded to a $1° \times 1°$ grid in
order to calculate MMM and MCM; results for individual models are illustrated based on the original grids. The MC regional
averaged value in this paper is calculated from region of 25°S-25°N and 80°E-170°E.

## 3   Results

In section 3.1, model results are compared to proxy reconstructions and the Hadley Centre sea ice and sea surface temperature
reanalysis version 1 (HadISST1) data set to evaluate model performance for reproducing mPWP and modern climate. In
addition, the multimodel mean method (MMM) is evaluated in comparison to individual models. We illustrate the most relevant
patterns of mean climate for the MC in mPWP relative to the preindustrial in terms of SST, precipitation minus evaporation,
wind stress and salinity at the ocean surface (section 3.2). Then we analyse oceanic circulation in the MC (section 3.3). In
section 3.4, we introduce a new metric – the multi cluster mean (MCM), which takes the weight of models from the different
clustering branches into consideration to better illustrate the results from multiple models. Finally, we present a comparison
between the performance of the multi-cluster mean (MCM), the MMM and individual models.

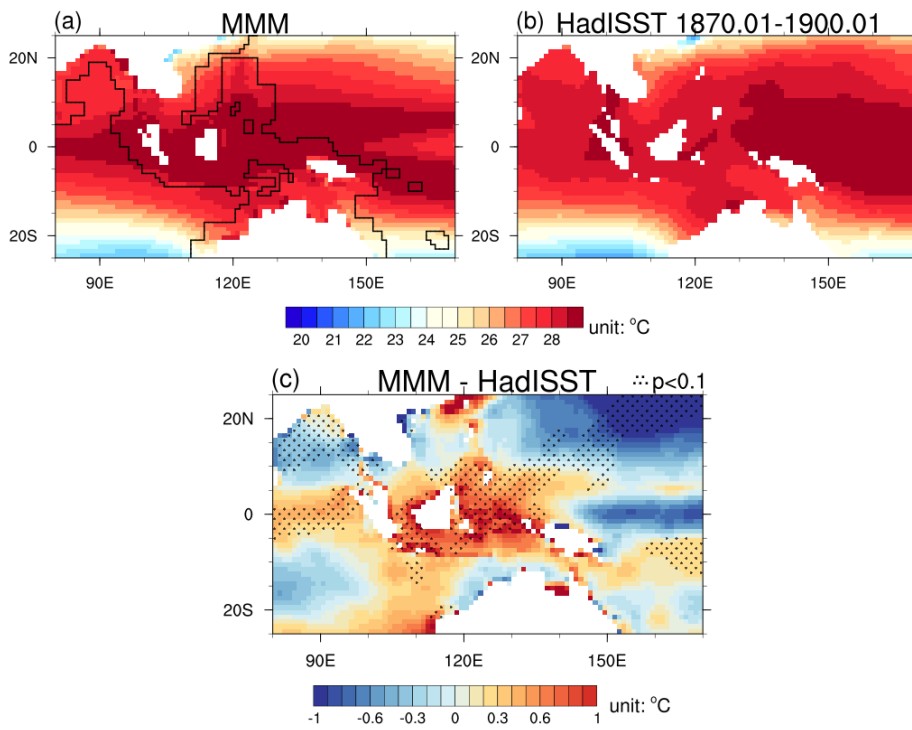

**Figure 2.** The annual mean sea surface temperature (SST) over the MC from: (a) PlioMIP2 multimodel ensemble mean (MMM) for the pre-industrial (E280), (b) HadISST1 reanalysed SSTs between January 1870 to January 1900 and (c) their difference. White shading indicates the grid boxes that are land for all the models. In (a), grid boxes outside the black lines are ocean grid boxes for all the models. Stippled areas show significant differences according to an one sample t-test at the 90% confidence level.

## 3.1  Model Skill at Simulating preindustrial and mPWP MC climate

In this section, we investigate how well a model performs in reproducing preindustrial climate (section 3.1.1) and mPWP climate (section 3.1.2). Our aim is to address the following questions: 1. Are the models that perform well in simulating preindustrial climate of the MC also good at simulating mPWP climate of the MC? 2. Does the multimodel ensemble have improved performance for the MC compared to individual models, as is often the case in weather and climate prediction (e.g.Tebaldi and Knutti (2007))?

### 3.1.1 Comparison with HadISST1

Observational data and reconstructed climatic information from proxy data can offer a way to evaluate simulation results. Here, we adopt HadISST1 data set (Rayner, 2003) from January 1870 to January 1900 to evaluate model performance in reproducing preindustrial climate. The HadISST1 dataset is available at https://www.metoffice.gov.uk/hadobs/hadisst/.

Fig. 2 shows the spatial distribution pattern of the preindustrial SSTs over the MC from the HadISST1 reanalysed SSTs and from the PlioMIP2 MMM. The MMM reproduces the SST zonal gradient from the equator to higher latitudes, but simulated SSTs are too warm surrounding the tropical islands and too cold in the North Western Pacific Ocean, compared to the reanalysis data. By calculating the root mean square error (RMSE) between reanalysed SSTs and model SSTs over the MC, we quantify the model and observation discrepancy for the modern period, which is shown in Fig. S1 and Table S1 in the supplement. The mean discrepancy in the MC between the MMM SSTs and HadISST1 SSTs is 0.51°C, which is lower (better) than 13 out of 17 models. By that metric, the MMM better reproduces preindustrial climate in this region than most of the individual PlioMIP2 models. The MMM metric also shows better agreement with observations than individual simulations in many other studies. For example, Williams et al. (2022) show that for the Deep-Time Model Intercomparison Project (DeepMIP) model ensemble, the MMM is the best estimate of the present day precipitation in terms of spatial patterns relative to individual DeepMIP models.

In terms of individual models, CCSM4 shows the lowest discrepancy of any individual model (0.39°C); IPSLCM5A and HadCM3 show the highest discrepancies (1.29°C). In order to evaluate the simulated SSTs' spatial patterns, the discrepancy between the MMM SSTs and the HadISST1 SSTs is also calculated after bias adjustment (i.e. removing the mean SST over the MC). The values are as shown in Figure S1 in the supplement.

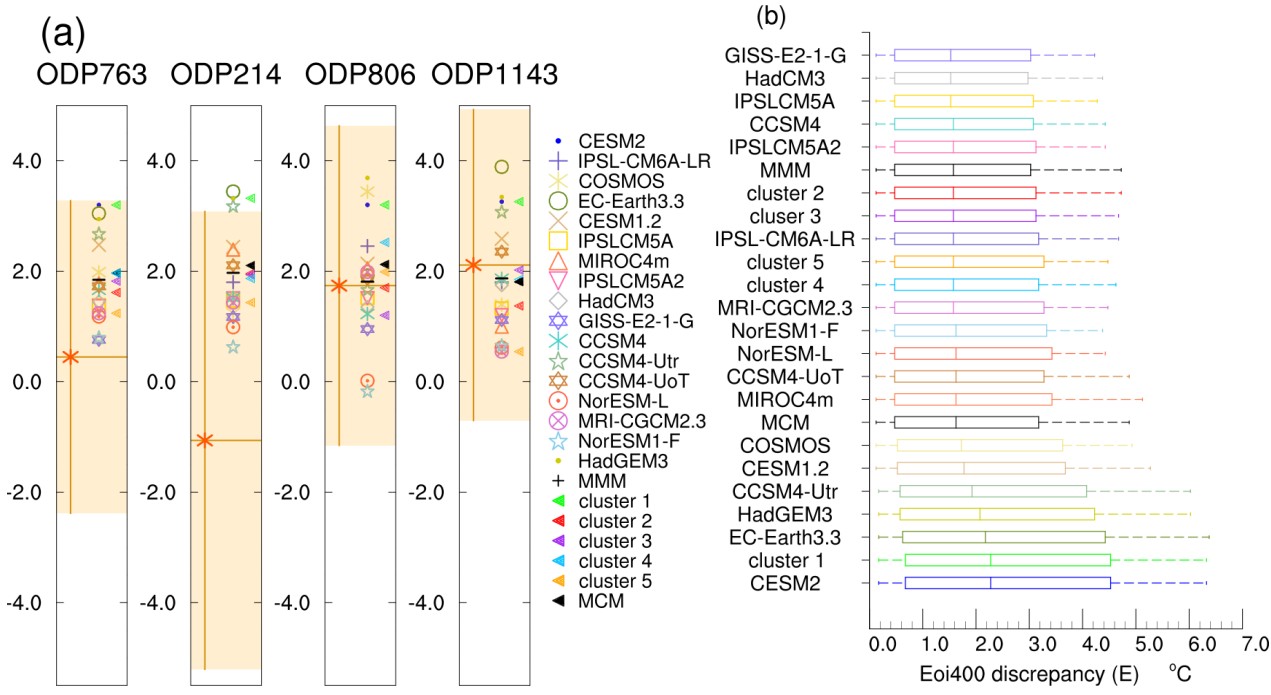

**Figure 3.** Discrepancy between modelled and reconstructed SST anomalies (SSTA) for the mid-Piacenzian warm period (mPWP). (a) Comparison between simulated and reconstructed SSTAs for 4 sites in the MC. The location of these 4 sites can be seen in Fig. 4a. The orange vertical bars and shading indicate the double standard errors ($\pm 2\sigma$) of the reconstructed SSTs. The orange horizontal bars and the asterisk indicate the reconstructed SSTAs of the site, individually. The short black horizontal bars among models' results indicate the MMM of the simulated SSTAs. Triangles refer to the results by different clusters (colored) and to the MCM (black). (b) Box-plot of model to proxy discrepancies over the MC for every model. Vertical lines inside the boxes are the median values (E) of the probability density function of the discrepancies derived via Monte-Carlo simulations, which is explained in section 3.1.2. Lower and upper box boundaries indicate the interval where the model and proxy discrepancies are *likely* to be (probability range from 66% to 100%). Lower and upper error bars indicate the interval where the model and proxy discrepancies are *most likely* to be (probability range from 90% to 100%). Models are ordered by their value of E. Also shown are the E values for the 5 clusters and the MCM.

### 3.1.2 Comparison with proxies

For the mPWP, multi-proxy reconstructed SST anomalies (SSTA) data compiled by McClymont et al. (2020) are available for a narrow time slice centred on 3.205 Ma BP. This data (McClymont et al., 2020) is compared with the simulated mPWP SSTA in this study. Data and the full details for reconstructed SSTA are available at https://doi.pangaea.de/10.1594/PANGAEA. 911847 and from McClymont et al. (2020). There are 4 sites with available data over the MC, as shown in Fig. 4a. Sites ODP214 and ODP763 are located at the eastern Indian Ocean; site ODP806 is located at the western tropical Pacific Ocean; site ODP1143 is located near the islands of the MC. The SSTA for ODP1143 is reconstructed from UK37 with the alkenone-derived (BAYSPLINE) calibration (Tierney and Tingley, 2018); the SSTA for ODP763, ODP214 and ODP806 are reconstructed from planktonic foraminifera *Trilobatus sacculifer* Mg/Ca with the Mg/Ca-derived (BAYMAG) calibration (Tierney et al., 2019a). Fig. 3a shows reconstructed and modelled SSTAs for every site individually. Model results for each site are provided as the average of the 9 grid boxes centred on the site in order to reduce local bias.

There are large standard errors ($\sigma$) for the proxy reconstructed SSTAs, with $\pm 2\sigma$ shown with the orange shading in Fig. 3a. Models align well with the reconstructed SSTAs at site ODP806 in the WPWP and at site ODP1143 in the South China Sea. For the sites ODP763, ODP806, and ODP1143, all the models' results are within the uncertainties of reconstructions. For the site ODP214, a few (two) of models suggest comparably warm mPWP SSTA that are outside the uncertainty range of the reconstruction. We note that there is a persistent mismatch between Mg/Ca-derived (BAYMAG) and alkenone-derived (BAYSPLINE) SSTAs. McClymont et al. (2020) show that generally and with few exceptions, Mg/Ca-derived SSTAs are lower than their alkenone-based counterparts. In particular at lower latitudes there is a substantial mismatch. For example, at site ODP1143, which has both UK37 and Mg/Ca data available, Mg/Ca suggests a negative SSTA, result from this proxy is not supported by alkenones. For the Mg/Ca data at sites ODP763, ODP214 and ODP806, the Mg/Ca-derived SSTAs are all cooler than the MMM SSTA. In particular, for site ODP763 and for site ODP214, all models simulate warmer SSTAs than the reconstructions, with a larger distance to the reconstructions for site ODP214. For this site, in the Indian Ocean the proxy data show the mPWP SST that is cooler than in the preindustrial, but all the models show warmer SSTAs. The cold biases of the Mg/Ca-derived SSTAs may be attributed to additional environmental factors, such as seasonality and the calcification depth of foraminifera (Hertzberg and Schmidt, 2013; Hönisch et al., 2013; Tierney et al., 2019b; McClymont et al., 2020), which can explain larger biases in the tropics where the thermocline is relatively deeper than at higher latitudes. Moreover, the foraminifera *Trilobatus sacculifer* adopted in ODP214 and ODP763 broadly derives larger negative SST anomalies compared to another surface-dwelling warm water foraminifera *Globigerinoides ruber* in the study of McClymont et al. (2020), which aligns with results from some studies (Fairbanks et al., 1980; Ravelo and Fairbanks, 1992; Curry et al., 1983) that in the tropics *Trilobatus sacculifer* is often found in a slightly deeper habitat than *Globigerinoides ruber* (Tierney et al., 2019b). Furthermore, studies such as Zhang et al. (2014) and Smith et al. (2020), show that reconstructed SSTAs from organic geochemical proxies, also suggest that the SSTA in site ODP806 and region near the ITF outlet could potentially be warmer.

In order to quantify discrepancies between proxy reconstructed and simulated SSTAs from individual models, we use Monte-Carlo simulations following a similar method to that used in Kageyama et al. (2021). The results are shown in Fig. 3b, sorted

in ascending order of model and proxy discrepancy median value E. Since there are uncertainties for proxy data, we generate an array containing 1000 random SSTAs using a uniform range distribution derived from the reconstructed SSTAs and $\pm 2\sigma$ for every site. Uncertainties of the simulated SSTAs are derived over the last 100 years of annual mean SSTAs. By subtracting the reconstructed SSTAs arrays from the 100 simulated SSTAs for every site separately and combining the arrays for the 4 sites, we derive arrays of model and proxy discrepancy for individual models which contain 400,000 (1000×100×4) SSTAs. A

box plot illustrating *likely* and *very likely* range of the probability density distribution function of the model and proxy absolute discrepancy is shown for every model in Fig. 3b. The vertical lines inside the boxes are the medians of the probability density distribution function, which means that there is a probability of 50% that model and proxy discrepancy are less than this value. The box in Fig. 3b shows the range of the *likely* model and proxy discrepancy; the dash error bar shows the range of the *very likely* model and proxy discrepancy. The likelihood definition adopted here is that from the IPCC Guidance Note (Mastrandrea

et al., 2010); *likely* means the probability of the outcome ranges from 66% to 100%; *very likely* means the probability of the outcome ranges from 90% to 100%. By using this method, we take uncertainties of proxy data and model simulation into account. From Fig. 3b we find that the E values between proxy and models range from 1.52°C to 2.28°C. As the majority of the model SSTAs are warmer than the reconstructed SSTAs (Fig. 3a), the models showing relatively less regional warming also show a lower discrepancy value E with proxy SSTAs over the MC than the models that produce higher warming (Fig. S2). The

PlioMIP2 models' MMM shows a discrepancy value E of 1.58°C, which is lower than that of most of the PlioMIP2 individual models.

In order to explore whether those models, that perform well in simulating preindustrial SSTs over the MC, are also good at simulating the mPWP SSTAs over the MC, the discrepancies of models and observed SSTs on the one hand and reconstructed SSTAs on the other hand are shown in Fig. S3 for both experiments. In terms of simulating the climate of the MC, there is

no clear correlation between model performance in preindustrial climate and in mPWP climate, implying that models, which reproduce preindustrial climate of the MC well, are not always good at simulating mPWP climate of the MC in accordance with the proxy reconstructions. However, we note that the uncertainty of discrepancies between model and reconstructed SSTA in the MC is significant due to limited sites for comparison and high uncertainties of the proxy reconstruction. From Fig. S3, models seem to fall into two groups: models where the discrepancy in the Eoi400 experiment appears decoupled from that in the

E280 experiment, and models where regional biases are linked between the E280 and Eoi400 simulations (including CESM2, EC-EARTH3.3, HadGEM3, NorESM1-F and CESM1.2). Models in the second group also appear to produce relatively warm in the Eoi400 simulation (Fig. S4). In terms of simulating the global SSTA, the comparison of the model and data discrepancies on global scale between the E280 and Eoi400 simulations is shown in Fig. S5, which shows a better correlation than the regional SST simulation.

**3.2   Mean Climate Features**

In this section, we describe the modelled mean climatic characteristics of the mPWP relative to the preindustrial over the MC in terms of SST, wind stress, P-E, and SOS.

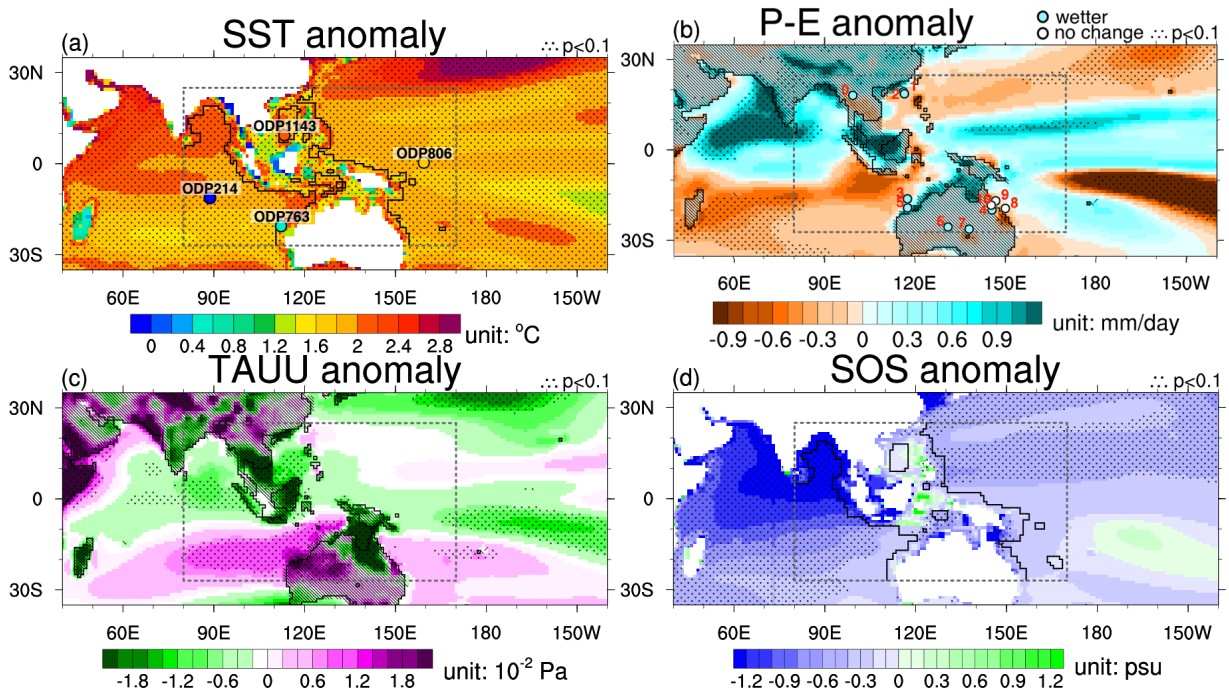

**Figure 4.** The MMM anomalies over the MC in the Eoi400 simulation relative to the E280 simulation for (a) sea surface temperature (SST), (b) precipitation minus evaporation (P-E), (c) zonal surface wind stress (TAUU) and (d) salinity at the ocean surface (SOS). In (a), circles mark the location of the SSTA proxies. Their colour represents the reconstructed SSTAs, the value of which is also shown in Fig. 3a, and the name of each site is indicated on the top of each circle. White shading and black lines in (a) and (d) are as in Fig. 2. In (b) and (c), hatches indicate land grid boxes in the Eoi400 experiment; the black lines indicate the continental outlines of the preindustrial experiment. In (b), filled circles denote proxy data in terms of wetter, or no-change condition. Details of each numbered site are shown in Table S2 in the supplement. In (c), green shading indicates westward anomaly and purple shading indicates eastward anomaly. Stippled areas show significant multi-model differences according to a t-test at the 90% confidence level.

The MMM shows a regional averaged warm SSTA of 1.88°C over the MC, the spatial pattern of which is as shown in Fig. 4a: the WPWP and the southern sector of the Indian Ocean in the MC the anomaly is less warm compared with other regions.

SSTAs are comparably warm over the North Indian Ocean in the MC region. Oceans surrounding the land masses of the MC also show apparently warm SSTAs, but this is not a robust signal because the models' land sea masks differ in the region of the MC. Models showing the greatest regional average warm SSTAs in the MC also show the largest global averaged warmings (Fig. S4). The linear regression coefficient between regional (the MC) and global annual SSTAs is ~1.06, implying that the SSTAs over the MC change at a similar magnitude as global SSTAs. The SSTA spatial patterns over the MC from individual

models are shown in Fig. S6.

Fig. 5 shows the correlation of equilibrium climate sensitivity (ECS) and MC SSTAs from models. Regarding the SSTAs, models consistently show warmer SSTs over the MC in the mPWP relative to the preindustrial; regional averaged SSTA range from 0.43°C produced by NorESM1-F to 3.33°C produced by HadGEM3. HadGEM3, EC-Earth3.3 and CESM2 all produce SSTAs exceeding 3°C, representing the 3 largest warm anomalies in the region among the PlioMIP2 models; 3 of the lowest

SSTAs, which are all less than 1°C, are from NorESM1-F, NorESM-L, and MRI-CGCM2.3, which are also the models with the lowest equilibrium climate sensitivity (ECS) among PlioMIP2 models (Haywood et al., 2020) as shown in Fig. 5. On long time scales, ECS is constrained by pattern effects i.e. radiative feedbacks change as SST patterns evolve (Ceppi and Gregory, 2017; Zhou et al., 2016; Armour, 2017; Gregory and Andrews, 2016; Dong et al., 2020; Andrews et al., 2015; Stevens et al., 2016; Zhou et al., 2021). For CMIP5 models, the WPWP is a key region, a warming in this region dominates the difference of

radiative feedback by enhancing Earth's albedo (Zhou et al., 2021; Dong et al., 2020). In the study of Haywood et al. (2020) which adopted CMIP6 and non-CMIP6 models, there is a relationship between models' ECS and their global temperature anomalies (Eoi400 minus E280). With the regard to warming of the WPWP in CMIP6 models, the MC contributes to the climate feedback differences, besides the effect of the MC, there is a stronger sensitivity of extratropical clouds to surface warming that may also result in differences (Dong et al., 2020). As such, the relationship of ECS and SSTAs on the MC is

not exactly linear. For example, GISS-E2-1-G shows an SSTA less than 1°C, but its ECS is relatively high. CCSM4-UoT and CCSM4-Utr have a modest ECS but show high global near-surface air temperature anomaly (Haywood et al., 2020) as well as a relatively high SSTA over the MC in mPWP. The best fit line of the linear relationship between ECS and SSTAs on the MC is ECS=2.43°C+0.74xSSTA at the 95% confidence level with $R^2 = 0.56$.

The warmer MC SSTs not only influence the atmospheric circulation in the mPWP but also bring higher evaporation due to

higher water vapour holding capacity of warm air (the Clausius-Clapeyron equation) and more atmospheric moisture, which contributes to the change of simulated precipitation. As an essential part of the hydrological cycle, the freshwater budget over land and ocean (P-E) influences ocean salinity and shapes the climate of the MC. Fig. 4 illustrates the MMM anomalies of P-E, zonal wind stress (TAUU) and SOS over the MC in the mPWP relative to the preindustrial.

For the MMM P-E anomalies, if we take the MC as a whole, the P-E is more positive in the mPWP over the MC relative to

the preindustrial (Fig. 4b); 13 out of 17 models and the MMM consistently show an increase in P-E (Fig. S7). Unlike SST, P-E does not change consistently over the MC but varies throughout the region (Fig. 4b), increasing over the Western Pacific Ocean and the North Eastern Indian Ocean, while decreasing over the west oceanic sector of the MC gateway. The difference in the

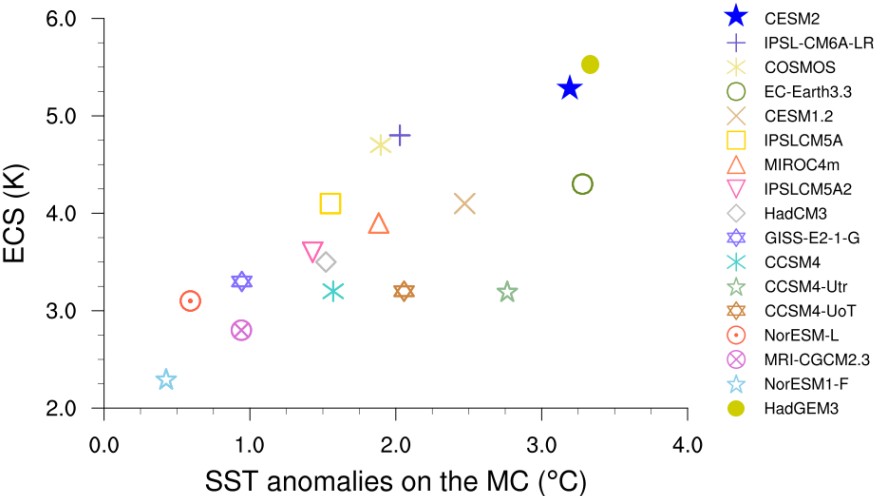

**Figure 5.** SST anomalies over the MC versus equilibrium climate sensitivity (ECS) from all the PlioMIP2 models.

distribution of land and sea in the Eoi400 and the E280 simulations is evident from hatching and black lines in Fig. 4b, which denote land grid boxes in the mPWP experiment and continental outlines of the preindustrial experiment, respectively. P-E

increases over most lands of the MC, and also increases generally over grid cells which are ocean in the preindustrial but land in the mPWP, leading to a land bias of precipitation increase. The spatial patterns of P-E anomalies from individual models are quite different (Fig. S7). In the case of some models, the simulated P-E anomaly is nearly opposite to each other, such as for COSMOS and EC-Earth3.3. Even the same family of models can show different results (e.g., CCSM-Utr and CCSM-UoT).

A large amount of precipitation provides energy to the atmosphere by releasing a large amount of latent heat, which fuels

the atmospheric circulation. In order to explore the changes of atmospheric circulation over the MC, we analyse the TAUU anomalies, as shown in Fig. 4c. Green shading indicates a westward anomaly. There are westward anomalies over the North Western Pacific Ocean and eastward anomalies over the ocean to the west of Australia. This means that the tropical trade wind has been enhanced on the side of the North WPWP during the mPWP. This signal can be seen from half of available PlioMIP2 models (Fig. S8). There are extreme values over some regions which belonged to a land mass during the mPWP but have turned

into ocean in modern times. The extreme values found there may therefore result from the change of land-sea distribution that also impacts on the prevalence of surface roughness. The latter impacts atmospheric flow more over land than over ocean.

As a result of the changes in the hydrological system, SOS changes surround the MC. Fig. 4d illustrates the spatial distribution of SOS anomalies. We find a decrease of SOS in most of the ocean regions belonging to the MC. This pattern is especially pronounced in the northern part of the Indian Ocean, where SOS decreases by more than 1.2 PSU. For the Pacific Ocean sector and the southern Indian Ocean sector of the MC, SOS decreases by less than 1 PSU. Overall, we find that seawater becomes fresher at the surface around the MC in the mPWP in comparison to the preindustrial. This finding is consistent in 14 out of 16 PlioMIP2 models and in the MMM. Spatial patterns of SOS anomalies for individual models are shown for reference in Fig. S9.

### 3.3 Ocean Currents

Temperature and density differences are factors that can contribute to changes in the ITF, which is an important process for regulating the climate of the MC. As the seawater characteristics of Pacific Ocean and Indian Ocean are so different, oceanic flow through the MC not only transports volume, but also creates a net-transport of salinity and heat between these two oceans. This process can directly influence both local and remote climate. Therefore, we quantify and investigate the changes of oceanic flow through the MC in the mPWP. In this section, we address the following questions: Is there more fresh and warm water injected into the Indian Ocean from the Pacific Ocean in the mPWP relative to the preindustrial? Where do the anomalies originate from? Do the anomalies result from the surface ocean, or rather from processes and characteristics that are shaped in deeper layers of the sea?

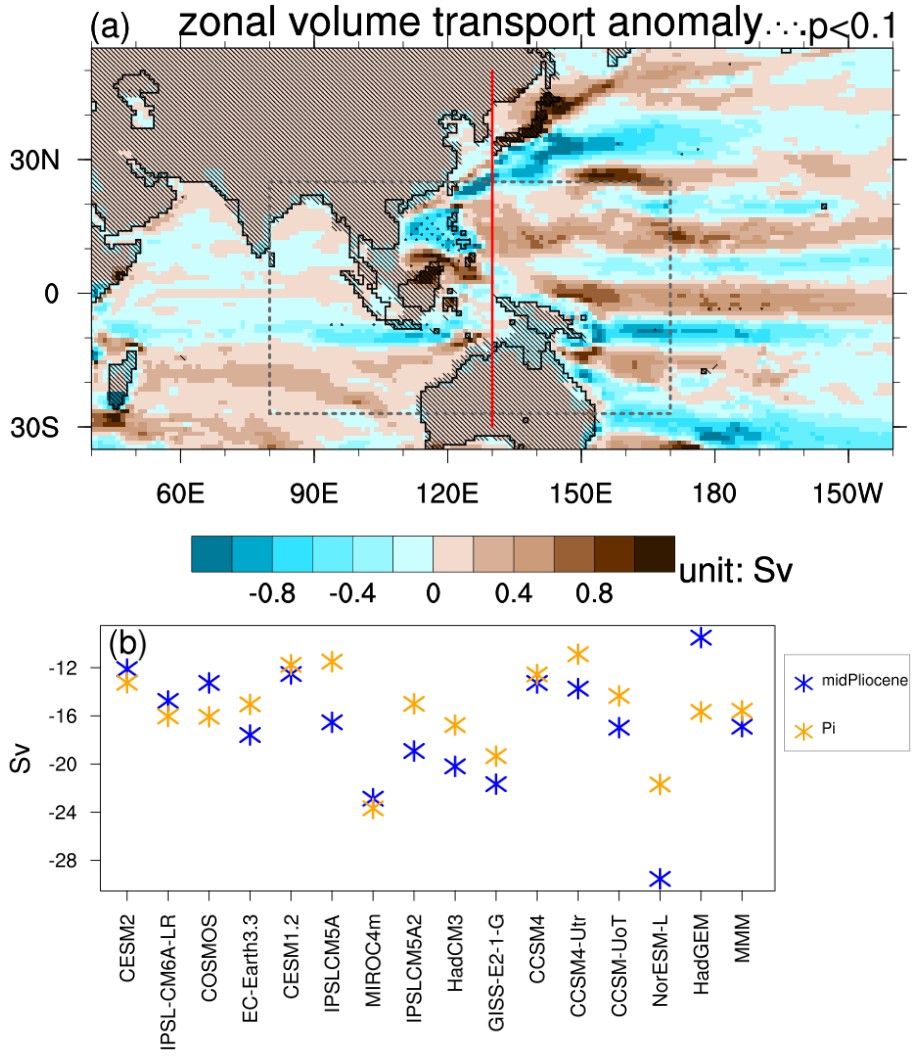

**Figure 6.** (a) The MMM zonal seawater volume transport anomaly in the Eoi400 experiment relative to the E280 experiment. Blue shading indicates westward anomaly and brown shading indicates eastward anomaly. The red line is the transect to calculate the ocean volume transport intensity in (b). The hatches indicate land grid boxes in the *mid-Pliocene-Eoi400* experiment; the black lines indicate the continental outlines of the preindustrial experiment. (b) The integrated ocean volume transport intensity through the gateway. Negative values indicate westward transport. Stippled areas show significant multi-model differences according to a t-test at the 90% confidence level. The maps of zonal seawater volume transport for individual models are as shown in Fig. S13.

We study the oceanic flow with the available data. During the calculation of ocean currents, we find that there are some disagreements between results derived from regridded data and analyses performed on individual models' native ocean model grids. Therefore, to reduce errors and improve the accuracy, we use native grid data to calculate the oceanic flow intensity. In order to include all the seawater transport through this gate and keep consistence for all the models, we calculate the seawater volume transport through a broad transect as shown with the red line in Fig. 6a. In the modelled mPWP, some regions, which are islands in the preindustrial, are still a peninsula that is connected to the Asian mainland in some models (CESM2, EC-Earth3.3, CESM1.2, MIROC4m, HadCM3, GISS-E2-1-G, CCSM4, CCSM4-Utr and CCSM-UoT). In these models except for HadGEM3 which adopted the same land-sea mask of the MC in both experiments, the Malacca Strait, and some other small straits in the northern part of the MC, are closed as shown in Fig. 1. Except for this gateway, the strait between New Guinea and Australia is also closed. Therefore, the Sunda and Sahul shelves caused some straits of the modern MC, that are present in the E280 experiment, to be absent in the Eoi400 experiment.

The spatial pattern of the vertical integrated zonal seawater volume transport anomaly is shown in Fig. 6a. There is a westward transport anomaly along the northeastern coast of the New Guinea, which favours the ocean mass transport of the ITF. For the oceanic transport through the gateway, the westward water transport via Timor passage has been strengthened over the north, but weakened over the south in the mPWP compared to the preindustrial. Models and modern observation both show a strong flow through the gateway between MC and Australia. According to the observation between 2004 and 2006 (Sprintall et al., 2009), the ocean volume transport via the Indonesian Throughflow to the Indian Ocean is 15 Sv. By integrating the water volume transport through this gateway vertically, we derive the total seawater transport volume through the MC, which is shown in Fig. 6b. Based on the model results, the flow through the MC ranges from 10.9 Sv (CCSM4-Utr) to 23.7 Sv (MIROC4m) westward in the preindustrial and from 9.5 Sv (HadGEM3) to 29.5 Sv (NorESM-L) westward in the mPWP. The result of the E280 experiment from every model shows the same transport direction and a magnitude comparable to observations. Compared to the preindustrial, the strength of the flow shows an increase in 10 out of 15 models, ranging from 5.6% (CCSM4) to 43.8% (IPSLCM5A). Five reductions are observed in the simulation by CESM2, IPSL-CM6A-LR, COSMOS, MIROC4m and HadGEM3. As mentioned before, the gateway geometry in HadGEM3 is the same in both E280 and Eoi400 experiments, which may lead to a result different from those of the other models. Therefore, 10/15 of the PlioMIP2 models show agreement on an increase in ocean volume transport through the MC in the mPWP in comparison to the preindustrial (NorESM1-F and MRI-CGCM2.3 are absent). Five of 15 models (CESM2, IPSL-CM6A-LR, COSMOS, MIROC4m and HadGEM3) disagree with the rest of the ensemble on the sign of transport change.

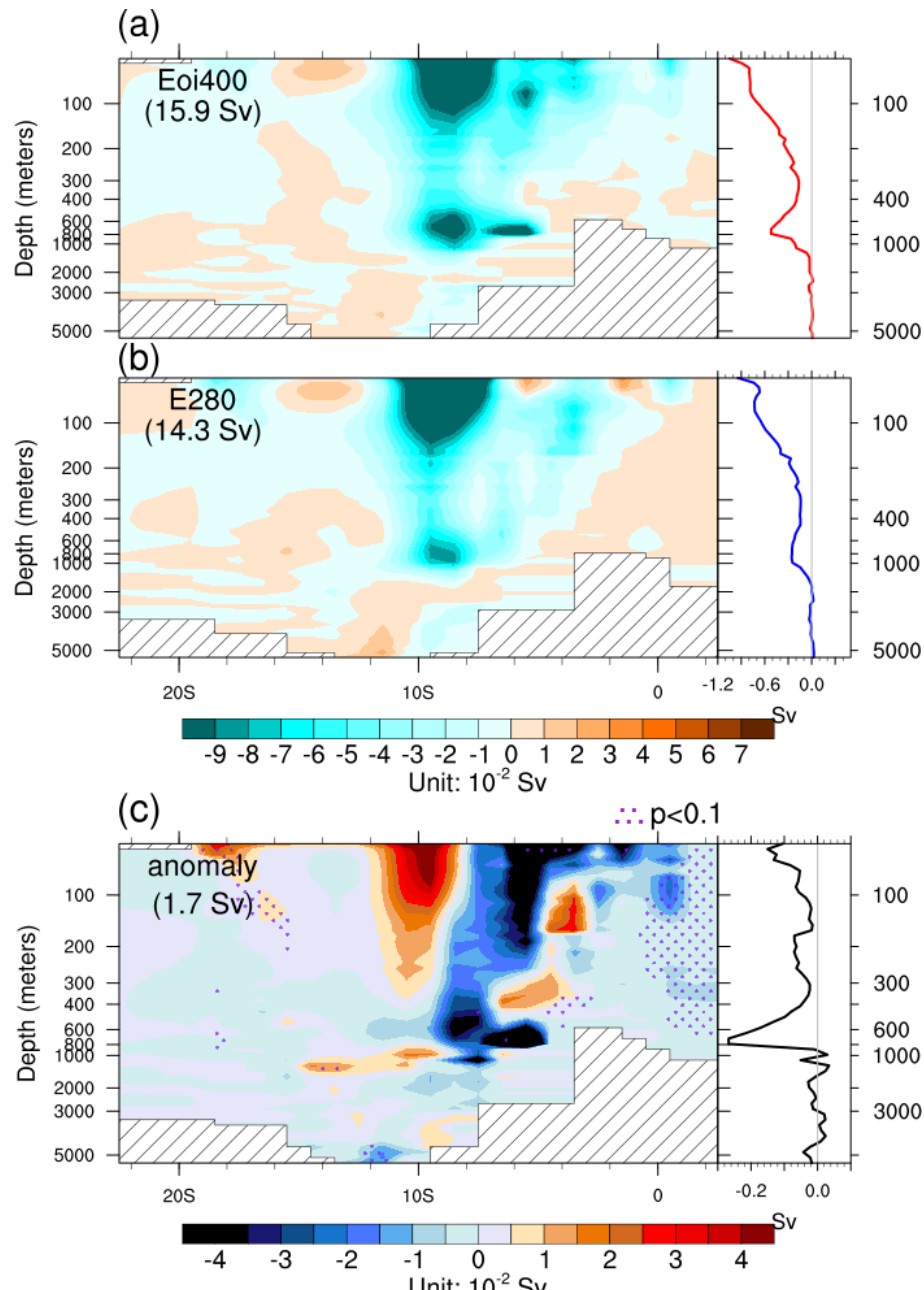

**Figure 7.** The MMM profiles of the ocean volume transport through the MC in the mid-Piacenzian warm period (mPWP) (a), in the prein-
dustrial (b) and the anomalies of the mPWP relative to preindustrial (c). The graphs attached on the right show the horizontally averaged
ocean volume transport. Values in brackets indicate the total water volume transport through the gateway; positive values indicate westward
transport. Warm shading indicates direction from west to east. Cold shading indicates direction from east to west. The upper colour bar is
for absolute values in (a) and (b). The colour bar on the bottom is for anomaly values in the (c). Stippled areas show significant multi-model
differences according to a t-test at the 90% confidence level.

In order to explore which depth of the ocean determines the changes of throughflow, we investigate the vertical structure of water volume transport through a broad passage of the ITF – Timor passage as shown by red line in Fig. 1, the MMM of which is shown in Fig. 7. In the MMM result, the strongest westward water transport is via the current located around 10°S. In general, the direction of water transport is the same in mPWP and the preindustrial. This encompasses the westward transport through the main ITF passage as well as the relatively weak eastward transport through the south branch of the gateway simulated by some models (pale yellow shading in Fig. 7). The strong westward flow counterbalances the weak eastward flow, and there is net-transport of fresh and warm water from the Pacific Ocean to the Indian Ocean. In both experiments, water mass is mainly transported in this gateway in the upper ocean above 1000 meters depth, which is consistent from the results of the MMM and individual models (Fig. S10).

The profiles of the throughflow anomalies are shown in the bottom panel of Fig. 7. From the difference between Eoi400 experiment and E280 experiment, it can be seen that the changes in meridional integrated ocean volume transport through Timor passage occur largely above 1000 meters depth, especially strongly at the surface and at a depth of around 800 meters, but without revolving the direction of water transport. In terms of the structure of the throughflow anomaly, there is a negative difference (stronger flow) in the northern part of the throughflow and a positive change (weaker flow) between 12°S and 8°S. This finding suggests that the flow through Timor passage developed closer to the northern boundary of the gateway in the mPWP than it is located during the preindustrial at the layers above 1000 meters.

### 3.4 Clustering

So far we have studied the climatic features of the MC by using the multimodel ensemble method, which has often been employed when analysing results from multiple models. However, we have shown that, although models largely show agreement on a warmer and wetter climate over the MC in the mPWP, the spatial patterns of climate anomalies vary between models (e.g. Fig. S6). These differences might counterbalance each other to some extent, and therefore potentially important climate signals from individual models will be eliminated when using the MMM. Moreover, some of the employed models belong to a group of general circulation models that share model components or may be similar in other respects. Consequently, similar signals from the same model family may be amplified in the ensemble of multimodel results. In order to investigate differences between models and to show the signals which might disappear in the MMM, we use clustering analyses in this section. Thereafter, we introduce a new metric, the multi cluster mean (MCM).

In the beginning of this section, we demonstrate the method of classifying models into groups based on the spatial features of SSTAs and precipitation anomalies over the MC from individual models by applying pattern correlation and the hierarchical clustering method, which are derived for example from Sierra et al. (2017) and Knutti et al. (2013a). After averaging the results from multiple clusters, we obtain the MCM results. Then we illustrate the spatial anomaly pattern of every group and of the MCM, and compare them with the MMM. After that, we investigate the performance of cluster groups and of the MCM at the example of the mPWP and the preindustrial simulation, and investigate if these show less discrepancy with observation/reconstruction than the MMM.

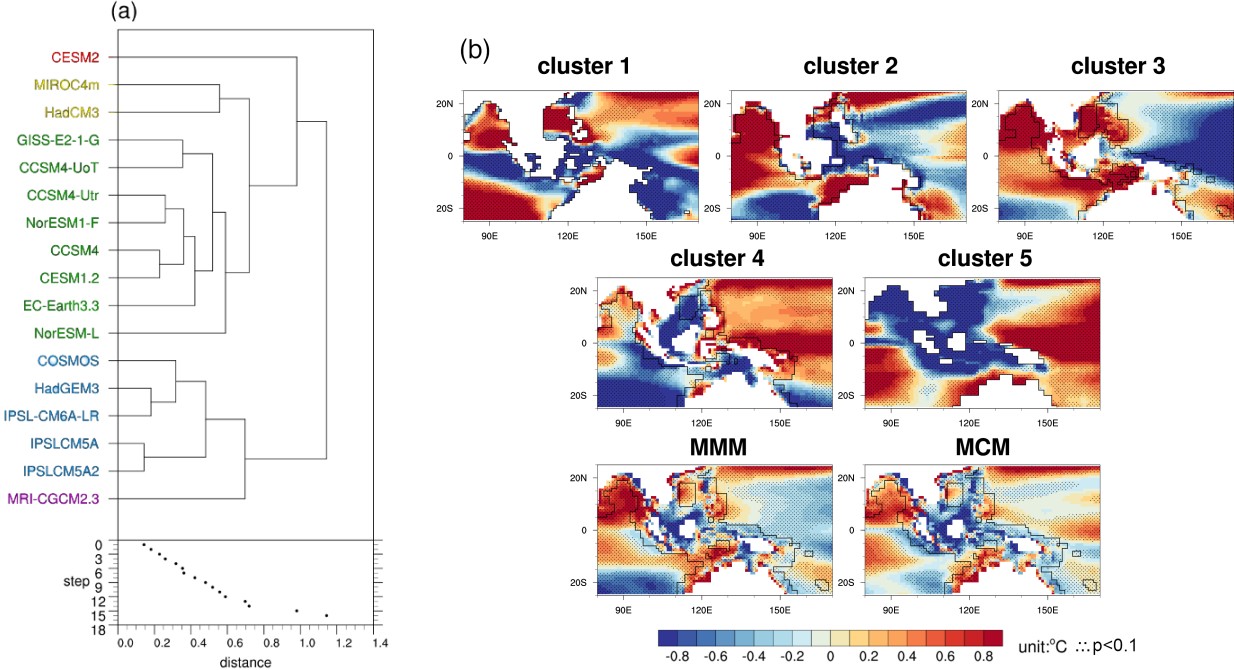

**Figure 8.** (a) Dendrogram of the SSTA clustering analysis based on the regional SSTAs over the MC from 17 PlioMIP2 models. Below the dendrogram the clustering step is shown as a function of distance between groups. (b) The SSTA patterns of the composite for different clusters, multi-model mean (MMM) and the multi cluster mean (MCM) after removing the regional mean SSTA. Cluster 1: CESM2; Cluster 2: MIROC4m and HadCM3; Cluster 3: GISS-E2-1-G, CCSM4-UoT, CCSM4-Utr, NorESM1-F, CCSM4, CESM1.2, EC-Earth3.3 and NorESM-L; Cluster 4: COSMOS, HadGEM3, IPSL-CM6A-LR, IPSLCM5A and IPSLCM5A2; Cluster 5:MRI-CGCM2.3. White shading indicates the grid boxes that are land for all the models of the cluster. The grid boxes outside the black lines are ocean grid boxes for all the models of the cluster. Stippled areas show significant differences according to a t-test at the 90% confidence level.

Fig. 8a and Fig. 9a illustrate the procedures of clustering. The first step is to find the two models with the smallest distance
and join them into one group. On each subsequent step, the two closest groups or models are merged into a larger group. The distance between two models is the pattern correlation between the multiannual mean SSTAs or precipitation anomalies of these two models. The x-axes of the Fig. 8a and the Fig. 9a indicate the distance of every two joined groups at every step. The pattern correlation measures the similarity of the spatial patterns between two models (Taylor, 2001), which is the Pearson Correlation Coefficient of the same variable of two different maps from two models. Here, for SSTAs we only use grid boxes
which are ocean in all the models. We choose average-linkage method (Wilks, 2011) to merge groups, which means the distance between two group is the averaged distance between individual members of the groups.

In the bottom panel of Fig. 8a, we find a large jump after clustering step 12, suggesting an appropriate stopping point with 5 groups, the group members of which are shown in Fig. 8a. CESM2 shows an SSTA spatial pattern that is different from that of the other PlioMIP2 models. Consequently it forms cluster 1 on its own. The same is true for MRI-CGCM2.3, but

this model could be merged into another cluster if we moved one step forward. Cluster 2 contains MIROC4m and HadCM3. Cluster 3 contains the largest number of models, with 8 members, including almost all the CCSM series model (except for CESM2), NorESM series models, GISS-E2-1-G and EC-Earth3.3. IPSLCM5A2 is an updated version of IPSLCM5A in terms of updated components and technical characteristics (Sepulchre et al., 2020). This is clearly shown in the clustering results, with IPSLCM5A and IPSLCM5A2 being the two closest models. All the 3 IPSLCM models have been classified into cluster

4 together with COSMOS and HadGEM3. Fig. 8b illustrates the ensemble SSTAs pattern of every cluster after removing the regional mean SSTA. Cluster 4 and cluster 5 show a relatively warmer pattern for the WPWP, which is opposite to results present in cluster 3. For the Indian Ocean, cluster 4 shows a relative weak warm pattern, which means the Indian Ocean warms less than the region, especially the WPWP. Spatial distributions of SSTA in the Indian Ocean are similar for clusters 3 and 2, warmer in the north but less warm in the south, which is opposite to results from cluster 5. Since models are equally weighted

in the ensemble, and cluster 3 contains the largest number of PlioMIP2 models, it dominates the MMM SSTA pattern (Fig. 8b). The MCM is the mean of these 5 clusters - in other words, it is the multimodel mean with weights. The MCM can avoid signals being overweighted from the same model family; it conserves details of different types of climate realization, that may come from similar or very different models. For example, the centre of the WPWP shows a warmer anomaly, a result consistent for all clusters except for cluster 3. Comparison between warming of the WPWP and the eastern Indian Ocean on different clusters

indicates 3 possible SSTA patterns: the eastern Indian Ocean warms more than the WPWP (cluster 2, cluster 3 and the MMM); the eastern Indian Ocean warms less than the WPWP (cluster 4); there is no apparent difference in warming between these two regions (cluster 1, cluster 5 and the MCM).

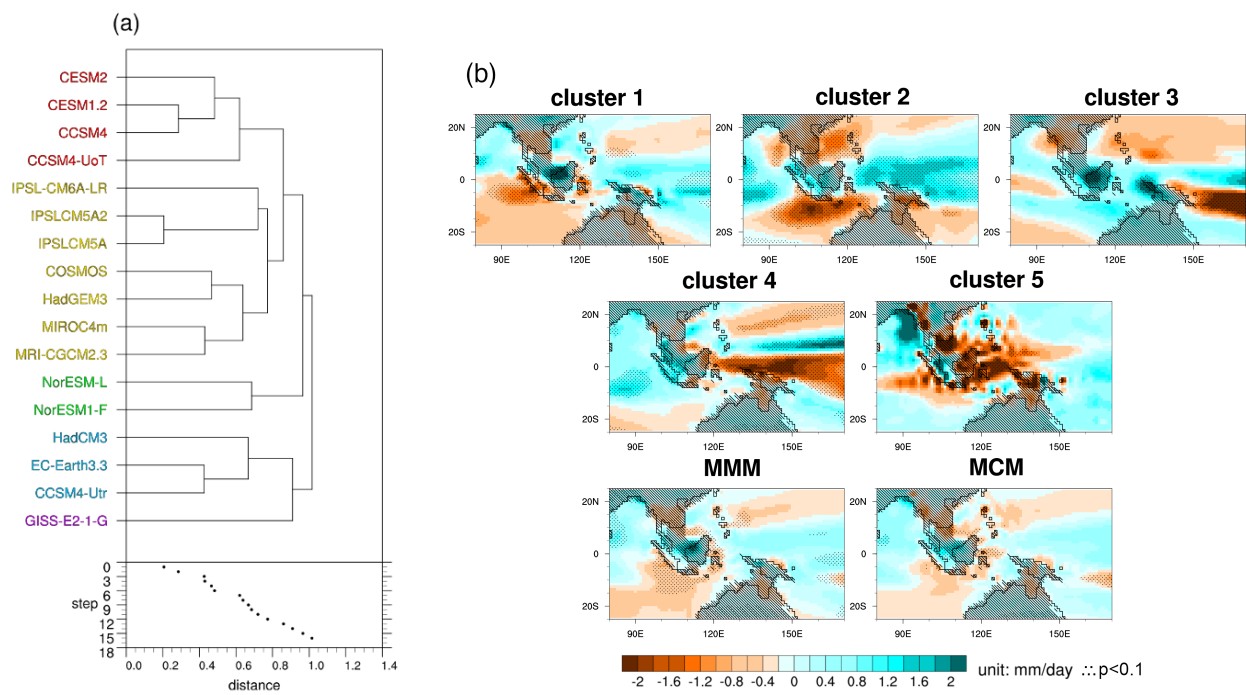

**Figure 9.** The same as Fig. 8 but for precipitation. Cluster 1: CESM2, CESM1.2, CCSM4 and CCSM4-UoT ; Cluster 2: IPSL-CM6A-LR, IPSLCM5A2, IPSLCM5A, COSMOS, HadGEM3, MIROC4m and MRI-CGCM2.3 ; Cluster 3: NorESM-L and NorESM1-F ; Cluster 4: HadCM3, EC-Earth3.3 and CCSM4-Utr ; Cluster 5: GISS-E2-1-G. The hatchings and the black lines are the same as Fig. 4. Stippled areas show significant differences according to a t-test at the 90% confidence level.

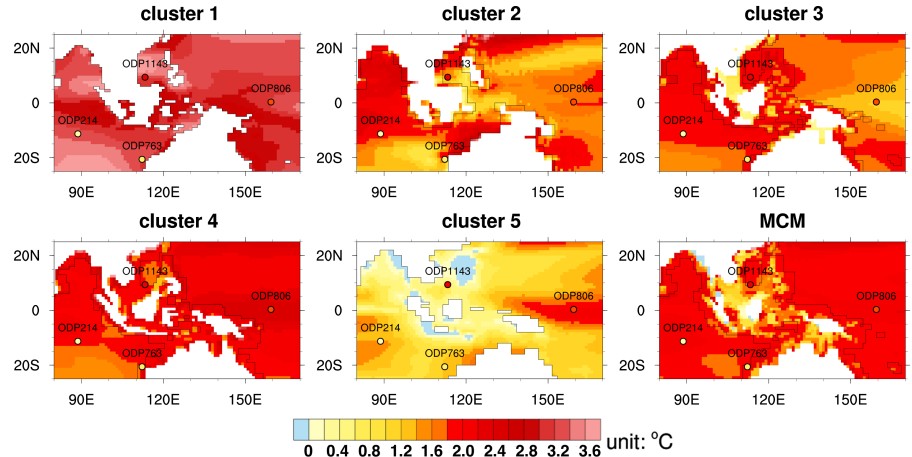

**Figure 10.** Comparison of reconstucted and modelled SSTAs over the MC in the mid-Piacenzian warm period (Eoi400) relative to the preindustrial (E280) from the SSTA-based clusters and the multi cluster mean (MCM). Cluster 1: CESM2; Cluster 2: MIROC4m and HadCM3; Cluster 3: GISS-E2-1-G, CCSM4-UoT, CCSM4-Utr, NorESM1-F, CCSM4, CESM1.2, EC-Earth3.3 and NorESM-L; Cluster 4: COSMOS, HadGEM3, IPSL-CM6A-LR, IPSLCM5A and IPSLCM5A2; Cluster 5:MRI-CGCM2.3. Units: °C. The MMM SSTAs shown in Fig. 4a. Note that in this figure, regional averaged SSTA has not been removed as in Fig. 8b. White shading and black lines are the same as in Fig. 8b

In addition to the cluster based on SSTAs, we also sort models based on the precipitation anomaly spatial patterns, as shown in Fig. 9. Similar to SSTA analyses, we find a jump after step 12. Therefore, we classify the models into 5 groups (Fig. 9a).

Four of the CCSM series models, CESM2, CESM1.2, CCSM4 and CCSM-UoT, form cluster 1. The IPSLCM-type models are members of the cluster 2 together with COSMOS, HadGEM3, MIROC4m and MRI-CGCM2.3. This cluster is the biggest one with 7 models. Similar to clustering based on SSTA, IPSLCM5A and IPSLCM5A2 are still the two most similar models; they are also the closest group for IPSL-CM6A-LR. NorESM-L and NorESM1-F form cluster 3. The remaining model in the CCSM series, CCSM4-Utr, joins EC-Earth3.3 and HadCM3 as cluster 4. For precipitation, cluster 5 (containing the single

model GISS-E2-1-G) shows a decrease in precipitation over the centre of the MC. For the WPWP, cluster 1 and cluster 2 show relatively wetter anomalies; cluster 3 and cluster 4 show basically drier anomalies with spatial heterogeneity. Whereas Cluster 5 (GISS-E2-1-G) alone shows a relatively drier climate in the northwest but wetter conditions in the rest of the WPWP. All clusters except cluster 5 show a relative dry anomaly over the north of the WPWP. Similar to SSTA-based clustering, the cluster with the largest number of models dominates the spatial pattern of the MMM and leads some signals of other clusters

to disappear. Overall, compared with the eastern Indian Ocean, the WPWP is relatively wetter (cluster 1, cluster 2, the MMM and the MCM), drier (cluster 3 and cluster 4), or receives similar amounts of precipitation (cluster 5). If we compare Fig. 8b with Fig. 9b, the pattern of the MCM is more similar to the pattern of the MMM in precipitation than in the SSTA.

Fig. 10 compares the 5 clusters with the reconstructed SSTA data. In order to quantify model clusters' performance in simulating the mPWP SSTAs, the SSTA-based clusters are validated in the same way as individual models and as the MMM, as shown in Fig. 3 and Fig. S3. The performance of the MCM is similar to the MMM for every site. However, in contrast to the MMM, the MCM can take the model weight into account to avoid duplication of model signals from the same family, and also retain the climate signals we might lose in the multimodel ensemble. An example is the warm anomaly signal in the centre of the Western Pacific Ocean.

## 4 Discussion

### 4.1 Physical Mechanisms

The MC is a region of intense atmospheric-oceanic coupling and exchange, where slight changes can affect climate afar (Gordon, 2005b; Yoneyama and Zhang, 2020). Due to differences in land distribution, topography, and $CO_2$ concentration, we find that the MC experienced warmer SSTs, higher freshwater budget, lower SOS, strengthened surface zonal wind in WPWP and increased oceanic volume transport through the mPWP compared to the preindustrial.

In the mPWP, wind and large scale atmospheric circulation change in response to an overall warmer climate. According to the PlioMIP2 simulations, the Northern Hemisphere warmed more than the Southern Hemisphere (Haywood et al., 2020). As a result of the variation of interhemispheric meridional heat fluxes and energetic constraints for the Intertropical Convergence Zone (ITCZ) position, the Southern Hemisphere Subtropical High pressure system is intensified, and the western Pacific easterly trade is enhanced correspondingly (Pontes et al., 2021), which is consistent with the strengthened zonal wind stress we find in Fig. 4c. In terms of atmospheric circulation, the Walker Circulation consistently shows a westward shift within PlioMIP2 models, but the change in the strength proved to be highly model dependent (Han et al., 2021). In contrast, PlioMIP1 models suggested a weakened Walker Circulation in the mPWP in relative to the pre-industrial (Corvec and Fletcher, 2017).

On multi-decadal timescales, the Pacific trade wind system plays a role on the circulation strength and patterns within the MC and Indian Oceans (Sprintall et al., 2014; Vecchi and Soden, 2007). This has been observed in the 1970s when the weakening of the trade winds resulted in the western Pacific shoaling thermocline anomalies (Vecchi et al., 2006; Wainwright et al., 2008), and this anomaly can be transmitted to the Indian Ocean by the ITF (Alory et al., 2007). Results from coupled models also reveal that the strength of the ITF reduced correspondingly with the weakened Pacific trade winds (Sprintall et al., 2014). On the projected anthropogenic warming world, most models suggest a reduction in the equatorial trade winds and a decreased ITF transport (Sen Gupta et al., 2012). In our study, most models show an increased ITF transport and also suggest an intensified western equatorial Pacific zonal wind stress (Fig. S11). The wind can influence water properties, thermocline and sea level through wave processes and therefore impact the formation of the WPWP (Wijffels and Meyers, 2004; Drushka et al., 2010; Pujiana et al., 2013; Sprintall et al., 2014). Changes in the WPWP have a direct effect on the ITF, as the ITF is driven by the sea-level gradient between the western Pacific (high sea surface height) and the eastern Indian Ocean (low sea surface height) (Wyrtki, 1987; Sun and Thompson, 2020; Gordon, 2005a). On decadal timescales, the Pacific Ocean sea surface height (SSH) variability controls more than 85% of ITF variation according to ocean reanalysis data (1959–2015) (Shilimkar

et al., 2022). Here, we also correlated the annual ITF volume transport with annual SSH by using data available from 4 models (Fig. S12). We find that generally the western Pacific SSH is positively correlated with the ITF. Furthermore, the SSH gradient between the western Pacific and the eastern Indian Ocean correlates well with the ITF in both E280 and Eoi400 experiments of GISS-E2-1-G and Eoi400 experiments of CESM2 and IPSL-CM6A-LR, but it has insignificant correlation with the ITF in Eoi400 experiment of EC-Earth3.3 as shown by values on the top right of Fig. S12. Moreover, the regions where SSH correlates with the ITF, differ between experiments and models. For CESM2, for example, experiments Eoi400 and E280 show different spatial patterns of the correlation coefficient map. However, models that show increased ITF in the Eoi400 do not always show an increase in SSH gradient and vice versa, implying at least for these four models, changes in the SSH gradient are not robust drivers for changes in the ITF between the mPWP and the preindustrial.

The ITF can transfer the signal of ENSO to the Indian Ocean and further afield. As shown in a study of ENSO from PlioMIP2, models GISS-E2-1-G, COSMOS, IPSLCM6A and CESM2 show an 'El Niño-like' mean state, while CCSM4-Utr, EC-Earth3.3 and CCSM4 show a 'La Niña-like' mean state in the mPWP compared to the modern Pacific Ocean (Oldeman et al., 2021). All the models that show a clear 'La Niña-like' mean state also show a stronger Pacific Walker Circulation (Figure 10a from Han et al. (2021)) and corresponding intensified ITF; IPSLCM6A, CESM2, COSMOS and HadGEM3 show 'El Niño-like' mean states at weakened ITF. Although GISS-E2-1-G shows an 'El Niño-like' mean state and unspecific zonal wind stress changes over the western tropical Pacific Ocean, it produces a stronger ITF. The rest of the models (IPSLCM5A, CESM1.2, HadCM3, NorESM-L) with intensified ITF, and MIROC4m with slightly weakened ITF, show a slight 'La Niña-like' mean state; IPSLCM5A2 and CCSM-UoT with intensified ITF show a slight 'El Niño-like' mean state (Oldeman et al., 2021) (Figure 4c).

The ITF as a component of the global ocean circulation system also interacts with other circulations. In PlioMIP2, most models simulate a stronger Atlantic Meridional Overturning Circulation (AMOC) in the mPWP than the preindustrial (Zhang et al., 2021b; Weiffenbach et al., 2023). On the centennial scale, the ITF decreases if the AMOC slows down as a northward surface flux anomaly implies a strengthened convergence of ocean water masses and increases SSH of the Indian Ocean (Sun and Thompson, 2020). Yet, as discussed for SSH above, the connection between changes in AMOC and ITF remains to be studied. Nevertheless, in the study of simulated future $CO_2$ induced warming world, changes of the overturning circulation in the Pacific basin explain the change of the ITF (Sen Gupta et al., 2016). The northward flow of deep waters entering the Pacific basin at 40°S is weakened, and associated net basin-wide upwelling to the north of the southern tip of Australia decreases, ultimately reducing the ITF (Fig. S10 in Sen Gupta et al. (2016)). In the mPWP, the Bering Strait is closed and water cannot be transported through this gateway and thus contribute to the increased ITF volume transport. In contrast, the volume transport through south Pacific basin and contribution to the ITF can be checked (Fig. S16). We find that all models that show decreased ITF volume transport show decreased volume transport through south Pacific basin (shallow or deep) in the mPWP relative to the preindustrial. For models showing increased ITF volume transport, the closed Bering Strait plays a role as the water leave the Pacific Ocean decreased, except for the HadCM3, which shows insignificant water transport via Bering Strait in the preindustrial. All models with increased ITF volume transport but GISS-E2-1-G show positive anomaly of the volume transport

through south Pacific basin above 2000 meters in the mPWP relative to the preindustrial, and it can contribute to the increased ITF volume transport.

The ITF is both impacted by and has an impact on the state of the surrounding oceans (Auer et al., 2019; Brierley and Fedorov, 2016; Gordon and Doherty). By comparing simulations with the MC gateways closed and opened, the removal of the ITF causes warmer SSTs in the tropical Pacific and cooler SSTs in the southern Indian Ocean (Lee et al., 2002). Moreover,

simulations where the ITF is blocked also show an eastward shift in atmospheric deep convection and precipitation in response to the warmer Pacific SSTs (Sprintall et al., 2014). In the mPWP, the substantially increased western Pacific SST results in higher atmospheric moisture content (Allen and Ingram, 2002), which provides substantial water vapour that contributes to increased precipitation over the MC as part of the "wet-gets-wetter, dry-gets-drier" response. As a consequence of multiple factors, the spatial precipitation anomaly pattern forms as shown in Fig. 4b (Feng et al., 2022).

Precipitation minus evaporation is the largest freshwater component entering the ocean (Byrne et al., 2016). Increased fresh water flux provides a way to explain the general decrease of the SOS over the MC, especially the decreased SOS in the Northern Indian Ocean. The changes of topography also play an important role. The closure of the gateway between New Guinea and Australia can change the water source from the saline south Pacific to the less saline north Pacific (Cane and Molnar, 2001), which can cause the Northern Indian Ocean SOS to decrease significantly. Furthermore, higher precipitation and wetter climate

over the Sunda shelves, India, south slope of the Tibetan Plateau and the surrounding regions (as shown in Fig. 5b of Haywood et al. (2020) and Fig. 1b of Feng et al. (2022)) may increase the local run-off, which can also inject more fresh water into the ocean and then dilute salinity.

Large scale flow patterns seem to be simulated unambiguously across the PlioMIP2 model ensemble. In particular, 10/15 models agree on increased westward ocean volume transport through the MC during the mPWP and 5/15 models show de-

520 creased volume transport.

## 4.2 Summary of the PlioMIP2 Model Performance

While MC SST are significantly warmer in the mPWP, it seems that within the PlioMIP2 model ensemble MC climate is to some extent decoupled from model ECS (Fig. 5). Nevertheless, we find that models with a large ECS also produce rather warm mPWP conditions around the MC; and models with a small ECS show relatively cooler conditions (Fig. 5). When using models

to predict climate of the MC for different levels of $CO_2$ or changes in earth surface conditions, it is important to note that the models that best reproduce a recent regional climate of the MC as quantified by a comparison to modern observations are not necessarily the same models as those that produce the best agreement between simulated and reconstructed mPWP SSTAs (Fig. S3). Yet, at global scale, there is a correlation between model performance on simulating mPWP and modern climate (Fig. S5). From that point of view, the mPWP proves to be a real test bed for the skill of models for climate states that strongly

differ from the recent one.

In general, we find agreement between proxy reconstructions of SSTA and mPWP climate simulations. At ODP Site 214 a small number of models fall out of the uncertainty range of the reconstruction at the warm end. Yet, we note that ODP214 indeed appears somewhat cool in comparison to nearby SSTA reconstructions in the region. This leaves us with the possibility

that the models are not able to capture climate mechanisms that shaped the mPWP climate recorded at site ODP214. An alternative explanation for the discord at this site may be sought in the proxy data itself or in its interpretation. We note that the model-data comparison is hampered by large uncertainty ranges of reconstructed SSTAs and by limited availability of suitable SSTA reconstructions in the region. This highlights the value of more regional MC SSTA reconstructions for the mPWP. We also find that the simulated climate in PlioMIP2 has a much smaller uncertainty range than the reconstructed climate, as seen by the spread of simulated SSTAs across the PlioMIP2 model ensemble which is much smaller than the uncertainty bars attributed to reconstructions (Fig. 3a). This suggests that the MC proxy reconstructions are shaped by additional aspects that the models are not responsive to.

## 4.3 Model Hierarchy and Clustering

The multimodel ensemble mean (MMM) has frequently been adopted to illustrate modelling results. We found the MMM produces smaller discrepancy than more than half of the PlioMIP2 individual models in both mPWP and preindustrial climate simulations (Fig. S3). In the case of model independence, the model biases will be partially cancelled, which results in the MMM outperforming individual models (Solomon et al., 2007; Knutti et al., 2010). Yet, some models, such as models adopted in Coupled Model Intercomparison Projects phase 6 (CMIP6), are not fully independent from each other and might have similar biases (Abramowitz et al., 2019). Boé (2018) quantified the relationship between CMIP5 model components replication and proximity of their results, and found there is a clear relationship at both global and regional scales. Model clustering is a way to study proximity of models results' spatial patterns and to weight models to reduce reduplication of similar biases from the same model family.

In PlioMIP2, as in CMIP6, not all models are independent from each other. There are models that have been developed from the same source, with shared codes or using different generations of the same components as shown in Table 1. A study from Ho et al. (2016) adopted outputs from 41 CMIP5 models and clustered models according to the concept of "model genealogy" (Knutti et al., 2013b) and found difference between the weighted and unweighted multi-model ensemble mean. Yet, general conclusions from both methods did not change. In this study, we cluster models into groups according to the proximity of their results in simulating SSTAs and precipitation (section 3.4). Although some models used the same ocean component, they turned out to be in different SSTA-based cluster groups, such as CESM2. Descendants of CCSM4, CCSM4-Utr, CCSM4-UoT, CESM1.2, CESM2, NorESM-L and NorESM1-F, are all in cluster 3, except for CESM2 (Fig. 8a). Compared to the other CCSM4 models, CESM2 has updated the POP ocean component not only with respect to the parameterization but also with regard to the use of new schemes (Danabasoglu et al., 2020). Moreover, CCSM4, CESM1.2 and CESM2 have very different atmospheric components. In the study of Pacific Walker Circulation, CCSM4 showed strengthened Walker Circulation, while both CESM1.2 and CESM2 showed weakened Walker Circulation (Feng et al., 2020b). Nearly all the models that include the NEMO ocean component (EC-Earth3.3, IPSLCM5A, IPSLCM5A2, IPSL-CM6A-LR and HadGEM3) except for EC-Earth3.3, appear in the SSTA-based cluster 4 (Fig. 8a). In the analysis of El Niño variability from PlioMIP2 (Oldeman et al., 2021), the 'La Niña-like' mean state model group and the 'El Niño-like' mean state group are not the same as the SSTA-based clusters found in our study. This implies that clustering depends on the region and on the studied physical quantity. In terms

of atmospheric component, some models used the same atmosphere component but haven't been clustered into the same group based on precipitation. For example, CCSM4, CCSM4-UoT, CESM1.2 and CESM2 are all in cluster 1, but CCSM4-Utr is not in this group. The two closest models in both SSTA-based and precipitation-based clusters are IPSLCM5A and IPSLCM5A2. They adopted the same atmospheric component and oceanic component, as well as vertical and horizontal resolutions. Compared with the IPSLCM5A, IPSLCM5A2 updated model components and retuned the cloud radiative forcing (Tan et al., 2020), but both models still produce a similar climate signal over the MC. From the discussion above, we conclude that the model configuration can provide a reference to cluster models, but even models of the same model family may still produce different climatic signals depending on the analysis region or the studied climate characteristic. These results suggest that there is not always a clear trace of model family apparent via the clustering.

Moreover, the multimodel mean will lead to the loss of climatic signal (Knutti et al., 2010). In our study, there are noticeable variations between spatial anomaly patterns from individual models. By illustrating models' results in cluster groups, climatic signal loss can be reduced (e.g., the warm anomaly signal in the centre of the Western Pacific Ocean; Fig. 8b).

## 5   Conclusions

The subaerial Sunda and Sahul shelves and relative high atmospheric $CO_2$ concentration, combined with other forcings, lead to a different climate of the MC in the mPWP than in the preindustrial. In line with the global climate, all the 17 PlioMIP2 models (including HadGEM3) show higher SSTs in the mPWP MC, ranging from +0.43°C to +3.33°C, with a MMM warming by 1.88°C; 13 out of 17 models show fresh water flux P-E increased in the MC, the anomaly ranging from +0.04 mm/day to +0.50 mm/day. In terms of atmospheric circulation, the easterlies over the western tropical Pacific Ocean show enhanced intensity in the MMM; there is anomalous eastward wind stress over the ocean near Western Australia. Seawater salinity affects the thermohaline circulation; 14 out of 16 models (MRI-CGCM2.3 is absent here) show a lower SOS in the MC.

Even though the topography of the MC in the mPWP acts as a barrier for volume transport between the Pacific Ocean and the Indian Ocean compared to the preindustrial, 10 out of 15 models show the ITF intensity increasing by 5.6% to 43.8%. CESM2, IPSL-CM6A-LR, COSMOS, MIROCm4 and HadGEM3 show a decrease in intensity of the ITF, the amplitude of westward flow is here lowered by 3.4% to 39.4%. In terms of spatial differences, the MMM shows that the ocean volume transport of the Timor passage is stronger in the north but weaker in the south in the mPWP relative to the preindustrial. Moreover, the water mass is mainly transported through the ITF in the ocean above 1000 meters depth in both the preindustrial and the mPWP. The changes in the volume transport of ITF also mainly occur in the ocean above 1000 meters depth.

The models' performance has also been quantified in this work by comparing the discrepancies between model results and both reanalysis data and reconstructed proxy data. The results imply that models, which reproduce modern climate well, are not always good at simulating the mPWP climate of the MC according to the proxy reconstructions. Although, at global scale, there is a correlation between model performance in terms of simulating the mPWP climate and modern climate (Fig. S6).

The comparison between individual models and the MMM suggests that the MMM could reproduce the preindustrial SST of the reanalysis better than 13 out of 17 models, and show less discrepancy with reconstructed SSTAs than 12 out of 17 models.

There is a big difference in the spatial distribution of anomalies between models. Because some models will amplify similar signals, and because some climatic signals from individual models will get lost in the MMM, a new metric - MCM has been introduced. It takes model families weight into account, provides a new perspective on the ensemble of multiple models' results

and demonstrates different climate states by means of clusters. The evaluation of the MCM shows that it outperforms 12 out of 17 models in simulating preindustrial climate and shows less or equal discrepancy with reconstructed mPWP SSTAs than 10 out of 17 models, which suggests that the MCM is a valuable method for exploring multimodel results.

*Code and data availability.* The PlioMIP2 model data used in this work is available from the PlioMIP2 database upon request from Alan M. Haywood (a.m.haywood@leeds.ac.uk). PlioMIP2 data from CESM2, EC-Earth3.3, NorESM1-F, IPSLCM6A, GISS2.1G and HadGEM3 can

be obtained from the Earth System Grid Federation (https://esgf-node.llnl.gov/search/cmip6/). The boundary conditions for the PlioMIP2 experimental design can be downloaded from https://geology.er.usgs.gov/egpsc/prism/7.2_pliomip2_data.html. Most of the analyses in this paper are processed with NCAR Command Language (NCL), some of the scripts use here have been shared in https://github.com/XinRenn/Xin_MM-cluster.

*Author contributions.* XR carried out the data process, wrote the manuscript, and led the paper. DJL and EH provided assistance during the

615 whole work. DJL and CS contributed to manuscript writing. AH, AAO, BOB, CJRW, CS, CG, DC, GL, JCT, LES, MAC, MK, MLJB, NT, QZ, RF, WLC, WRP, XL, YK, ZZ and AMH conducted the simulations and provided data. All authors provided comments and contributed to this paper.

*Competing interests.* The authors declare that they have no conflict of interest.

*Acknowledgements.* This research is funded by the European Union's Horizon 2020 research and innovation programme under the Marie

Skłodowska-Curie grant agreement No 813360 4DREEF and the NERC SWEET grant (grant no. NE/P01903X/1). Wing-Le Chan and Ayako Abe-Ouchi acknowledge funding from JSPS (KAKENHI grant no. 17H06104 and MEXT KAKENHI grant no. 17H06323) and computational resources from the Earth Simulator at JAMSTEC, Yokohama, Japan. Development of GISS-E2.1 was supported by the NASA Modeling, Analysis, and Prediction (MAP) Program. CMIP6 simulations with GISS-E2.1 were made possible by the NASA High-End Computing (HEC) Program through the NASA Center for Climate Simulation (NCCS) at Goddard Space Flight Center. Charles J. R. Williams

acknowledges the financial support of the UK Natural Environment Research Council (NERC)-funded SWEET project (research grant no. NE/P01903X/1), as well as funding from the European Research Council (under the European Union's Seventh Framework Programme (FP/2007-868 2013) (ERC grant agreement no. 340923 (TGRES)). GL and CS acknowledge institutional funding at AWI via the research program PACES-II of the Helmholtz Association. CS acknowledges funding via the Helmholtz Climate Initiative REKLIM. MB and AvdH

acknowledge support by the program of the Netherlands Earth System Science Centre (NESSC), financially supported by the Ministry of Education, Culture and Science (OCW). Xiangyu Li acknowledge funding from National Natural Science Foundation of China (42005042 and 42275047). All IPSL simulations have been run on Très Grand Centre de Calcul du CEA (TGCC) through GENCI (Grand Equipement National de Calcul Intensif) allocation gen2212. We acknowledge the work of GENCI and TGCC for making our simulations available for the present work. We acknowledge the World Climate Research Programme, which, through its Working Group on Coupled Modelling, coordinated and promoted CMIP6, as well as the Paleoclimate Modelling Intercomparison Project (PMIP) for coordinating its fourth phase, including the mPWP simulations analysed here, with CMIP6. We thank the climate modeling groups for producing and making available their model output, the Earth System Grid Federation (ESGF) for archiving the data and providing access, and the multiple funding agencies who support CMIP6 and ESGF.

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
