# Peer review of "The hydrological cycle and ocean circulation of the Maritime Continent in the mid-Pliocene: results from PlioMIP2"

_EGUsphere, 2022_

## Author Comment (AC1)

First of all, thank you so much for helping us to improve our manuscript with your suggestions.

Our point-to-point responses are as below.

The original review comments are shown in blue, and our responses in **black**.

**Major Comments:**

✓ Results on ITF are not well connected with the rest of the manuscript. In other words, why should we care about the ITF in the Pliocene simulations (considering that we do not have proxy data to provide sufficient constraints on the model results)? In the current form of the manuscript, ITF is described separately from the SST and hydroclimate variables. Although, in the introduction, the author did cite literation on how the ITF is linked to coupled ocean-atmosphere variability and how the ITF may influence the monsoons. However, the authors results on ITF do not make any of the connection or mechanistic analysis. Given this disjoint, I am wondering whether the author should consider cutting the ITF results and focus on the regional SST and hydroclimate over the Maritime Continent instead.

**A: After consideration we still plan to to keep the results on the ITF into this study. The ITF is a factor that can influence and be influenced by the rest of climatic factors described in the manuscript. Regarding the disjointedness, we will try to integrate these sections better by putting a paragraph in the beginning of each section highlighting the linkages. Moreover, we will also add correlation analyses on the ITF strength to some variables such as temperature gradient across the Indian and Pacific Ocean, salt gradient and zonal wind strength and add more figures and analysis in the manuscript.**

✓ In the Discussion (Section 4.3), the authors stated that "but even models of the same model family may still produce different climatic signals depending on the analysis region or the studied climate characteristic." Can you provide explanation for this interesting result? Is it because of the potentially different model resolution, or details of the boundary condition implemented by different authors, or internal variability?

**A: In the revised paper we will explain the differences between, for example, CESM2 and other CCSM4 models with different parameterization and schemes. Although some models can be categorized into the same family, they are not identical; they have differences in resolution, parameterizations, or model components. These differences can result in different climatic signals. As for the boundary condition implementation, all the models run the simulations with the same boundary conditions (except HadGEM3 run Eoi400 with different land-sea mask than other models), and so in our study it is likely less important for causing the differences.**

✓ Are there available proxies on the hydroclimate (precipitation /evaporation and sea surface salinity) and ITF in the region? If yes, please include results and discussion on these comparisons. If no, please state it explicitly in the manuscript (that there is no available proxy for

**A: We have collected published proxies that indicate the wet/dry conditions in the study region and we will include them in Figure 5b.**

✓ Please consider adding a summary of model-proxy comparison of SST in the abstract.

**A: We will insert text similar to that below, in the abstract:**
By comparing model results with data it has been found that models, which reproduce modern climate well, are not always good at simulating the mid-Pliocene climate anomaly of the MC. In addition, the MMM reproduces the preindustrial SST of the reanalysis better than most individual models, and produces less discrepancy with reconstructed SSTAs than most individual models in the MC.

**Minor comments:**

✓ Lines 23–25: Rewrite and change into "A large amount of rainfall releases large quantities of latent heat into the atmosphere, which is an important driver of global atmospheric circulation".

**A: We will rewrite this line.**

✓ Many of the multi-panel plots are not labeled with subplot label (such as (a) and (b)). Please check and make sure all the subplots are properly labeled.

**A: We will check all the subplots and label them.**

✓ Information should be provided on how the ocean salinity was initialized in the simulations. This information is needed because the authors examined the sea-surface salinity changes in the PlioMIP simulations (e.g., Figure 5d), and it is not clear whether the ice-volume effect has been accounted for in the simulations and has an imprint in Figure 5d.

**A: The initial conditions of ocean salinity are either derived from (Levitus and Boyer 1994), an equilibrium state of the modern (control) simulation, or the end of the PlioMIP1 experiment (Haywood et al. 2011). Ice sheets have been accounted as boundary conditions in the experimental design. But, since ice sheet changes were prescribed, the PlioCore experiment will be in equilibrium with the ice sheets (Haywood et al. 2020). So the ice-volume effect won't have an imprint in the salinity. We will double check this with simulation groups at the revision stage, and we will include this information in section 2.2.**

✓ Line 266: "the relationship is not exactly linear."
**A: We will rewrite this line.**

✓ Figure 10: cluster 5 (GISS) looks weird. The model resolution is ~2 degree (Table 1). It is hard to believe the precipitation anomaly has such a rich fine structure. Please double check and make sure calculation has been done correctly.

**A: For visualisation of the clustered groups, we regridded all the individual models'**

**results into 1x1 degree so that we can combine models with different resolution.**

**Reference:**

Haywood, A. M., H. J. Dowsett, M. M. Robinson, D. K. Stoll, A. M. Dolan, D. J. Lunt, B. Otto-Bliesner, and M. A. Chandler, 2011: Pliocene Model Intercomparison Project (PlioMIP): experimental design and boundary conditions (Experiment 2). *Geosci. Model Dev.*, **4**, 571–577, https://doi.org/10.5194/gmd-4-571-2011.

Haywood, A. M., and Coauthors, 2020: The Pliocene Model Intercomparison Project Phase 2: large-scale climate features and climate sensitivity. *Clim. Past*, **16**, 2095–2123, https://doi.org/10.5194/cp-16-2095-2020.

Levitus, S., and T. P. Boyer, 1994: World Ocean Atlas 1994. Volume 4. Temperature. US Government Printing Office.

---

## Author Comment (AC2)

We appreciate it that you help us to improve our paper by giving your suggestions. We list our point-to-point replies as below.

The original review comments are shown in blue, and our responses in **black**.

This paper takes a careful, nuanced analysis of the potential biases in PlioMIP2 models' representation of mean climate state and whether this is related to bias in their simulation of Pliocene climate. This is important, since we normally justify the use of certain models to simulate past climates in a given region based on their ability to reproduce the pre-industrial climatology. However, if this is not the case, then we must be more careful of our choice of models. The results presented required a huge amount of data compilation and analysis, and it is a nice demonstration of the utility of the multi-cluster mean approach to understanding model disagreement and consensus.

**Thank you for your kind words.**

✓ Figure 3: make the markers larger, more bold, it's a little hard to see all of them directly. Would suggest using different colors rather than distinct symbols, or increase line weight or something. Make sure each panel is labeled (a) and b) do not appear on the figure

**A: As there are some overlapping, we agree that it does take some time to distinguish every marker. As such, we will make adjustments to this figure to make it clearer. Regarding to the panel label, we will check every figure and add labels to sub-figures.**

✓ Figure 4: is it possible to propagate through the uncertainty (e.g. full error envelope of Pliocene proxy values) into the EOI400 discrepancy calculations? Or at least add some 95% error bounds.

**A: Thank you for this useful comment - we agree that it is important to consider the uncertainties in the discrepancy (shown in Figure 3) when interpreting Figure 4. As such, we will add this to the figure caption and/or discuss in the text.**

✓ Figure 5, and generally all figures with continental outlines: make the outlines of the land bold.

**A: We will adjust the outlines of these figures.**

✓ Figure 6 is very interesting. The relationship between ECS and SSTA over the maritime continent - I wonder if this can be formally connected to the 'pattern effect' literature. See for instance:

Dong, Y., Armour, K.C., Zelinka, M.D., Proistosescu, C., Battisti, D.S., Zhou, C. and Andrews, T., 2020. Intermodel spread in the pattern effect and its contribution to climate sensitivity in CMIP5 and CMIP6 models. Journal of Climate, 33(18), pp.7755-7775.

This literature points to the fact that long-term changes in climate feedbacks seems to depend on the relative warming in the western Pacific warm pool region.

**A: Thank you - we will add appropriate discussion about the pattern effect in our discussion section 4.2.**

✓ While the text does a good job of articulating the differences between the MMM and the MCM, in the plots themselves they look quite similar for SSTA and precipitation. This might be a feature of the small size of the plots. Can the individual panels be made larger, and the dendrogram made much smaller, so that it is easier to compare and contrast the map panels?

**A: We will adjust the size of each figure.**

✓ Line 475: I would say more about the potential sources of proxy uncertainty for ODP 214. As of now the analysis of this potential source of model disagreement is quite cursory.

**A: We will expand the discussion in section 3.1.2, Line 202-209, of the underestimate of temperature reconstruction for the site ODP 214 which is Mg/Ca-derived and not be supported by other proxies such as alkenones.**

---

## Author Comment (AC3)

Thank you so much for your constructive comments and suggestions. It helps us to improve our manuscript a lot.

Here are our point-to-point responses.

The original review comments are shown in blue, and our responses in **black**.

**Major comments:**

✓ The analysis and methods used by the authors are not sufficient to achieve their central objective. First, the hydroclimate of the MC is very complex and is under the influence of changes in mean state features (such as Hadley and Walker circulation, ITCZ position and the warming itself) and important internal modes of variability (such as ENSO, IOD and IOBW). Secondly, the study focuses on a very small region around the MC, from which is not possible to obtain a picture of the large-scale dynamics. Many results are described in terms of the small scales changes around the MC (i.e., North-eastern coast of New Guinea, central MC, gateway between MC and Australia, etc), which are not possible to be evaluated from the coarse resolution of the PlioMIP models. Additionally, the PlioMIP models are required to apply substantial changes in the land-sea mask in the MC and are of coarse resolution (some up to 4 degrees in the atmosphere), making its small-scale evaluation very difficult and uncertain. Furthermore, it is not possible to infer how the MC hydroclimate was during the Pliocene by simply evaluating the basic fields described in section 3.2. I recommend the authors to expand their study area to include the Pacific and Indian ocean changes, as well as southern Asia, which is a large land mass above the MC, where any small temperature changes may affect the atmospheric circulation and ITCZ position. Also, to evaluate the MC hydroclimate the authors must show results of Hadley and Walker circulations and the possible influence of changes in the main modes of variability (ENSO and IOD).

**A: (a) Following your suggestion, we will expand our study area so that large scale information can be seen.**

**(b) We agree that these internal modes of variability are related to the mean state, which is the focus of this manuscript, so we will draw on results from other papers to discuss them more appropriately. In particular, some of these modes have been studied previously in other papers analysing the PlioMIP ensemble (e.g. Oldeman et al. 2021; Pontes et al. 2021). We will reference and discuss these papers where appropriate to provide more context to our analysis. In addition we will discuss changes in the Hadley and Walker circulations in our revised paper.**

**Regarding to the changes in the land-sea mask in the MC, the experiment of the mid-Pliocene does not only make changes in the land-sea mask in this small region but also other regions such as Bering Strait and Canadian Archipelago based on the reconstruction of this period. So the climate changes locally are not only forced by the local topography change but also changes in other regions**

✓ It is not clear how three of the result's sections (3.2, 3.3, 3.4) are linked to each other. These sections seem very independent from one another without a clear justification on why choosing these analyses to compose the manuscript. Section 3.2 must include more elements as mentioned above. In Section 3.3, the authors must show, through analyses, what the relative effect of an increased ITF in the mean MC hydroclimate is. After incorporating these new results,

the authors will evaluate the utility of performing a cluster analysis (comment #7). In order to provide a comprehensive story of the MC climate, each analysis must be clearly justified.

**A: We will integrate these sections more clearly and put a paragraph in each section to show the linkage between different sections and how the results of each section feeds in the other sections. In addition, we have decided to reorganize our sections, to make the linkages clearer, in particular we are considering moving section 3.1 before section 3.4. Regarding to section 3.2 and section 3.3, we will also add analyses of the relationship between ITF strength and other variables such as temperature gradients across the Indian and Pacific Ocean, salt gradient, and zonal wind strength.**

✓ The authors have not performed any statistical significance analysis of the fields and processes evaluated. As such, it is not possible to know what the major changes simulated by most of the models are, and how these could be related to one another. Performing statistical significance analyses for each result is crucial before publication.

**A: We will add statistical significance indicators to all the Figures.**

✓ In section 3.1, I suggest removing the cluster results for Figure 3. At this point of the manuscript the authors have not provided enough information of the cluster analysis and can confuse the readers. The analysis shown in Figure 4 is not very elucidative. Comparing discrepancies is very uncertain, especially for the mid-Pliocene where proxy-data show larger uncertainties. Is there any precipitation record in the MC that the authors could compare the PlioMIP results to? Borneo?

**A: As stated above, we will move this section to later in the manuscript, in order to improve clarity.**
**Regarding to Figure 4, also in response to Tripti's online comment, we will add discussion to this section on the uncertainties associated with the discrepancy. In terms of the precipitation record, also in response to Reviewer 1, we will add a comparison with proxy precipitation data in this region.**

✓ In section 3.2, the authors analyse SST and P-E changes by averaging these variables over the study area. However, the study area encompasses the Indian and Pacific oceans as well as many artificial the occur due to a modified land-sea mask in the mid-Pliocene. As such, this is not a good metric to evaluate the MC climate.

**A: Thank you for this suggestion. In terms of SST and SOS, we agree that there are some potentially artificial changes due to changes in land-sea mask, and we will account for this in the regional means. However, for P-E the changes real, and potentially important for climate.**

✓ In section 3.3, the authors again try to address specific questions that are not possible to be addressed because either the study area is too restricted or beacuse of low resolution of the models. To address where the salt or heat anomalies originate from, it would be necessary to evaluate the large-scale heat and salt budget. It is not possible address whether the ITF anomalies originate from the surface or deep due to coarse resolution of the model and because

of the land-sea masks, which will likely have different effects in each model due their different resolutions. I suggest the authors to focus on the possible role of the ITF transport on the MC hydroclimate. I am also very confused on what is being shown on Figure 8. The legend says 'ocean current' but the unit is Sv, which is a unit of transport.

**A: We will expand our study region out to include the larger area that is associated with the Indonesian Throughflow, and will add some discussion on the relationship between the ITF and MC hydroclimate.**
**Regarding to the legend of Figure 8, thank you for pointing this out, we will change it to "ocean volume transport".**

✓   In my understanding a cluster analysis would be more appropriate if the PlioMIP models did not show a clear agreement on the changes for the MC climate, in which some models show very distinct results that could be masking important simulated features. However, without statistical significance analyses it is not possible to evaluate the usefulness of the cluster analysis. Also, the authors argue that the cluster analysis would reduce the influence of models of the same family in the MMM results, but it is not shown that models of the same family produce similar results. In fact, I found some results very different. SSTs from CESM2 are very different from all other CESM models. CCSM4-UoT is quite distant from CCSM4 and CESM1.2. NorESM models are several steps apart. Finally, the authors say that 'the MCM can avoid signals being overweighted from the same family of models, but one could argue that the MCM could also vanish changes that are simulated by most models. For example, cluster 3 of SST includes nearly half of the models analysed.

**A: We will add the statistical significance analyses to the cluster analysis.**
**As for the reduction of similar bias from models of the same family. This is a good point that we will add discussion to the paper.**
**Regarding to the MCM, the cluster analysis is most useful when the ensemble mean is made up form models that show a wide range of different results, with some grouping within these. We will make this clearer in the paper, and will add an explanation the number of group members needs to be considered when using this method.**

✓   Discussion, conclusions and abstract must be rewritten on the light of the comments above.

**A: We will edit our manuscripts based on the suggestions we received.**

**Minor comments**

L 34: ITF must have units of transport and not temperature.

**A: This unit here is for the heat that ITF transported. In order to avoid mistake, we used the original text from the reference Sprintall et al. 2009.**

L 104: Review grammar of Q2 or rephrase it.

**A: We will rephrase it.**

L 107: Q4. It must be first demonstrated that there is a duplication of biases in the PlioMIP ensemble. Models' results from the same family can change quite substantially.

**A: We will add this context to the Q4.**

L135: The title of section 2.2 must be modified to not mislead the readers, once it is not performed any simulations specific to this study.

**A: We will change the title into 'PlioMIP2 experimental design". This study is based on the background of the mid-Pliocene and how the hydroclimate of the MC change with the global climate. Therefore, we will change the title so that readers can know they are experiments designed in the PlioMIP2 project.**

Figure 3: The location of the sites used to construct this figure must be plotted in figure 1, otherwise it is not conductive for a good reading.

**A: This land-sea distribution map here we show is constructed from Dowsett et al. (2016) which is retrodicted with a set of procedures based on the paleogeographic maps of Markwick(2007). The symbols of sites will cover the coastal lines which is important information we want to show in this figure. In order to make it clear, we will indicate this in the figure caption.**

Figure 4: Do you use the mean SST around the MC to plot this figure? If so, this is not appropriate for the proxy-data because there are only a few sites around the MC.

**A: Yes. For the discrepancy of the pre-industrial simulation with observation data we use the mean different around the MC, since that's how we quantify the performance of models in simulating pre-industrial. We make this figure to compare model performance in simulating the climate of the MC in these two periods. For mid-Pliocene simulation performance quantification, there are a few sites of reconstructed temperature around the MC. As such, we will discuss about the uncertainties associated with the discrepancy in this section when interpreting this Figure.**

L 245: You cite fig. 6, but fig. 5 has not been cited yet.

**A: Thank you for reminding us. I will manage this in the revised manuscript.**

L 266-268: What do you mean by 'linearity is not exactly linear'? This sentence seems to be confusing what is shown on the plot.

**A: We will change this sentence into "the correlation is not exactly linearly".**

L 284-285: this sentence needs a better theoretical explanation.

**A: We will rephase this sentence into: A large amount of precipitation provide energy to the atmosphere by releasing a large amount of latent heat, which fuels the atmospheric circulation.**

Figure 6 needs a statistical significance analysis with correlation coefficient and p-value.

**A: We will add statistical significance indicators to this figure.**

L 295: I suspect the SOS decrease in the North Indian Ocean maybe related to a northward shift of the ITCZ, which is an important feature of the MC climate.

**A: That's a good idea that we will test it and may add it to the discussion.**

L 301-304: This is a too simplicity view of the drivers of the ITF. The ITF is an important feature of the large-scale ocean circulation and is not driven by density gradients between the Pacific and Indian oceans.

**A: Thank you for clarifying it. We will change this sentence. Instead of talking about the drivers of the ITF, we will say the wind and density gradient are factors that can make contributions to the ITF.**

L 329: Results must be described with the assistance of statistical methods to quantify significance. Means and standard deviations? Medians and inter-quartile ranges?

**A: We will do the statistical significance analysis when calculate the water volume transport intensity. And we will show it in Figure 7b in the revised manuscript.**

L 351: Why it could be expected a reverse in the direction of the ITF?

**A: We are curious if the direction of the ITF ever changed in the paleo periods so in this sentence we point it out that the direction keeps the same as today.**

L 388: Why is it necessary to remove the regional mean SSTa in figure 9b?

**A: We cluster models based on the spatial pattern of SSTa without the influence of overall regional mean SSTa, as some of the models show much warmer mean SSTa than the other models. As such, we removed the mean regional SSTa.**

Figure 11: It is not possible to clearly see the colour corresponding to the proxy-data.

**A: We use this colour scale to fit the results from all the clusters and the proxy. In order to make it clearer, we will point out in the figure caption that the exact value is showed in Figure 3.**

Figures: All figures need statistical significance analyses in order to be more confident of the results described in the text.

**A: We will add statistical significance indicators to our results.**

**Reference:**

Dowsett, H. J., and Coauthors, 2016: The PRISM4 (mid-Piacenzian) paleoenvironmental reconstruction. *Clim. Past*, https://doi.org/10.5194/CP-12-1519-2016.

Markwick, P. J., 2007: The palaeogeographic and palaeoclimatic significance of climate proxies for data-model comparisons. *Geol. Soc. Spec. Publ.*, 251–312, https://doi.org/10.1144/tms002.13.

Oldeman, A. M., and Coauthors, 2021: Reduced El Niño variability in the mid-Pliocene according to the PlioMIP2 ensemble. *Clim. Past*, **17**, 2427–2450, https://doi.org/10.5194/cp-17-2427-2021.

Pontes, G., and Coauthors, 2021: Northward ITCZ shift drives reduced ENSO activity in the Mid-Pliocene Warm Period. *Nat. Geosci.*, **Preprint**, https://doi.org/10.21203/RS.3.RS-402220/V1.

Sprintall, J., S. E. Wijffels, R. Molcard, and I. Jaya, 2009: Direct estimates of the indonesian throughflow entering the indian ocean: 2004-2006. *J. Geophys. Res. Ocean.*, **114**, 7001, https://doi.org/10.1029/2008JC005257.

---

## Editor Decision (ED1)

**Minor recommended corrections**

*All line references refer to the version of the manuscript with tracked changes.*

Please consider the use of mid-Pliocene throughout the manuscript and whether it is stratigraphically correct. From Dowsett et al 2016:

The PRISM time slab or PRISM "interval", as defined above, occurs within the Piacenzian Age. The Piacenzian is roughly equivalent to the Gauss normal-polarity chron (~ 3.6 to 2.6 Ma). Prior to 2010, the Pliocene Epoch included the Zanclean, Piacenzian, and Gelasian ages. Thus, it was common practice to refer to the PRISM interval as the midPliocene. Changes enacted by the International Commission on Stratigraphy revised the placement of the Pliocene– Pleistocene boundary from the base of the Calabrian Stage (1.801 Ma) to the base of the Gelasian Stage (2.588 Ma) (Gibbard et al., 2010). This change makes it awkward to refer to the mid-Piacenzian PRISM interval as mid-Pliocene (see also Dowsett and Caballero-Gill, 2010). Previous publications referring to the PRISM interval, PRISM time slab, mid-Pliocene warm period (mPWP), or mid-Piacenzian all refer to the same interval of time originally defined by Dowsett and Poore (1991) and revised by Dowsett et al. (2010) as discussed above (Fig.1). **We propose the term mid-Piacenzian be used hereafter**.

Please note that Haywood et al 2020 also use the term mid-Piacenzian (3.264-3.025 million years ago) and the short form MP. It is acknowledged by the editor that a variety of ways of describing this period do exist in the literature, so this is not always straightforward, but where possible it should be kept consistent with Dowsett et al (2016) and Haywood et al (2020).

BP to denote before present is also perhaps unnecessary here as it refers specifically to radiocarbon dating with present being 1950.

Line 111 - what are the characteristics…

Line 117-119 - Wording is a little off. Please consider rephrasing second part of question: How can we deal with the effect of any bias duplication from models within the same 'family' within our analysis?

Could reference Table 1 on line 131 and remove sentence on 139 that says: Refer to Table 1 which shows the components used in the PlioMIP2 Models.

Line 161 - the map is quite small in the version of the manuscript online, it is not easy to see the ocean gateways - can you please make the left-hand map larger in the final figure upload.

Line 164 - there is still a ? Within the sentence, so it looks like there is missing information - please correct.

Line 200 - perhaps the word should be compare rather than confer?

Line 227 - For example, at site ODP1143, which has both UK37 and Mg/Ca data available, Mg/Ca suggests a negative SSTA mid-Pliocene versus preindustrial, result from this proxy is not supported by alkenones.
Please consider the phrasing here - I am not sure what is meant by a negative SSTA mid-Pliocene versus preindustrial. Perhaps you don't need mid-Pliocene versus preindustrial as this is inferred in the Anomaly part of SST?

Line 250: show that reconstructed….

Line 254: we use Monte-Carlo simulations following a similar method used in Kageyama et al. (2021).

Line 262: Here we use letter E to denote discrepancy. I think this is not needed as you already refer to the proxy discrepancy median value (E) in line 254.

What is the figure on page 3 of the supplementary information? It has no legend and doesn't look like it is connected to Fig. S1 or Fig. S2

Line 288 - I am not sure you have previously defined P-E or SOS.

Please check the clarity of figure 4 in the final manuscript carefully. I think the ODP labels and the site numbering for panel b might be difficult to decipher.

Line 301 - which representing the 3 largest warm anomalies needs correcting to either representing the 3 largest warm… or which represents the 3 largest warm….

Line 309-311 referencing the Dong et al 2020 paper would benefit from re-writing as I am not clear of the point that is being made.

Line 326: From hatching and black lines in Fig. 4b, the difference in the distribution of land and sea in the Eoi400 and the E280 simulations is evident. Would be useful if you refer to what the hatching and black lines denote here to help the reader.

Caption for Fig 7 - positive values indicate westward transport

Fig. 10 - please consider the scale on the colour bar

Line 507: Changes in the WPWP have a direct effect…

Line 526 - impact of ENSO on the Indian Ocean

Page 30 and Section 4.2 - there are five uses of on the one hand/other hand. Please consider alternative phrasing for this section.

Line 608: As in CMIP6 , also PlioMIP2, not all the models are independent from each other. - consider rephrasing to "In PlioMIP2, as in CMIP6, not all models are independent from each other".

Line 657: remove the 'on the one hand' and 'on the other hand' here. It is not needed.

---

## Author Response (AR2)

Thank you so much for your comments.

Our point-to-point responses are as below.
The original review comments are shown in blue, and our responses in **black**.

Please consider the use of mid-Pliocene throughout the manuscript and whether it is stratigraphically correct.

A: We agree on it and changed mid-Pliocene into mid-Piacenzian in this manuscript.

BP to denote before present is also perhaps unnecessary here as it refers specifically to radiocarbon dating with present being 1950.
A: We changed it into "3.264 to 3.025 million years ago" to keep consistent with Haywood et al (2020).

Line 111 - what are the characteristics…
A: We made the correction.

Line 117-119 - Wording is a little off.  Please consider rephrasing second part of question: How can we deal with the effect of any bias duplication from models within the same 'family' within our analysis?
A: We rephase it into "How can we account for the effect of any bias duplication caused by models from the same 'family' within our analysis?".

Could reference Table 1 on line 131 and remove sentence on 139 that says: Refer to Table 1 which shows the components used in the PlioMIP2 Models.
A: We removed sentence on 139 and put "(Table 1)" after the end of line 131.

Line 161 - the map is quite small in the version of the manuscript online, it is not easy to see the ocean gateways - can you please make the left-hand map larger in the final figure upload.
A: We updated this figure.

Line 164 - there is still a ? Within the sentence, so it looks like there is missing information - please correct.
A: It is a reference. We corrected this error.

Line 200 - perhaps the word should be compare rather than confer?
A: We deleted the unnecessary word "confer".

Line 227 - For example, at site ODP1143, which has both UK37 and Mg/Ca data available, Mg/Ca suggests a negative SSTA mid-Pliocene versus preindustrial, result from this proxy is not supported by alkenones.
Please consider the phrasing here - I am not sure what is meant by a negative SSTA mid-Pliocene versus preindustrial.  Perhaps you don't need mid-Pliocene versus preindustrial as this is inferred in the Anomaly part of SST?
A: We agree. We deleted "mid-Pliocene versus preindustrial".

Line 250: show that reconstructed….
A: We made the correction.

Line 254: we use Monte-Carlo simulations following a similar method used in Kageyama et al. (2021).
A: We rephased it into "we use Monte-Carlo simulations following a similar method to that used in Kageyama et al. (2021)".

Line 262: Here we use letter E to denote discrepancy. I think this is not needed as you already refer to the proxy discrepancy median value (E) in line 254.
A: We deleted this line.

What is the figure on page 3 of the supplementary information? It has no legend and doesn't look like it is connected to Fig. S1 or Fig. S2
A: We added caption to this figure.

Line 288 - I am not sure you have previously defined P-E or SOS.
A: We defined P-E and SOS in the end of the Introduction. For the track-changes file, it is Line 100.

Please check the clarity of figure 4 in the final manuscript carefully. I think the ODP labels and the site numbering for panel b might be difficult to decipher.
A: We added semi-transparent background for ODP labels. For site numbering in panel b, we bolded the label.

Line 301 - which representing the 3 largest warm anomalies needs correcting to either representing the 3 largest warm… or which represents the 3 largest warm….
A: We made the correction.

Line 309-311 referencing the Dong et al 2020 paper would benefit from re-writing as I am not clear of the point that is being made.
A: We rephased this sentence into "With the regard to warming of the WPWP in CMIP6 models, the MC contributes to the climate feedback differences, besides the effect of the MC, there is a stronger sensitivity of extratropical clouds to surface warming that may also result in differences".

Line 326: From hatching and black lines in Fig. 4b, the difference in the distribution of land and sea in the Eoi400 and the E280 simulations is evident. Would be useful if you refer to what the hatching and black lines denote here to help the reader.
A: We made this correction.

Caption for Fig 7 - positive values indicate westward transport
A: We made this correction.

Fig. 10 - please consider the scale on the colour bar
A: We redrew the scale on the colour bar.

Line 507: Changes in the WPWP have a direct effect…
A: We made this correction.

Line 526 - impact of ENSO on the Indian Ocean
A: We changed this sentence into "The ITF can transfer the signal of ENSO to the Indian Ocean and further afield."

Page 30 and Section 4.2 - there are five uses of on the one hand/other hand.  Please consider alternative phrasing for this section.
A: We deleted some unnecessary "on the one hand/other hand" and replaced some with other words.

Line 608: As in CMIP6 , also PlioMIP2, not all the models are independent from each other.  - consider rephrasing to "In PlioMIP2, as in CMIP6, not all models are independent from each other".
A: We made this correction.

Line 657: remove the 'on the one hand' and 'on the other hand' here.  It is not needed.
A: We made this correction.